# Frontopolar theta oscillations link metacognition with prospective decision making

Alexander Soutschek [1✉], Marius Moisa [2], Christian C. Ruff [2,3] & Philippe N. Tobler[2,3]

Prospective decision making considers the future consequences of actions and therefore requires agents to represent their present subjective preferences reliably across time. Here, we test the link of frontopolar theta oscillations to both metacognitive ability and prospective choice behavior. We target these oscillations with transcranial alternating current stimulation while participants make decisions between smaller-sooner and larger-later monetary rewards and rate their choice confidence after each decision. Stimulation designed to enhance frontopolar theta oscillations increases metacognitive accuracy in reports of subjective uncertainty in intertemporal decisions. Moreover, the stimulation also enhances the willingness of participants to restrict their future access to short-term gratification by strengthening the awareness of potential preference reversals. Our results suggest a mechanistic link between frontopolar theta oscillations and metacognitive knowledge about the stability of subjective value representations, providing a potential explanation for why frontopolar cortex also shields prospective decision making against future temptation.

[1] Department for Psychology, Ludwig Maximilian University, Munich, Germany. [2] Zurich Center for Neuroeconomics, University of Zurich, Zurich, Switzerland. [3] Zurich Center for Neuroscience, University of Zurich and ETH Zurich, Zurich, Switzerland. ✉email: alexander.soutschek@psy.lmu.de

Beliefs about our economic preferences guide our behavior no less than these preferences themselves: For example, only if we are aware of our health goals and our weakness for chocolate may we avoid the sweets shelf in the supermarket[1]. How precisely we can gauge our preferences is determined by our metacognitive abilities, which appear crucial for successful behavioral control given their contributions to symptoms of several psychiatric disorders[2–4]. In value-based choice, our metacognitive abilities allow us to judge how confident we are in our preferences. While numerous studies investigated how such economic preferences are represented in the brain[5,6], less is known about the neural mechanisms that guide value-based decision making by metacognitive judgments of preferences. The psychological literature conceptualizes metacognition as a construct that allows explaining the accuracy of introspective confidence reports, but for a long time it has been debated whether metacognition represents also a natural kind at the neural level[7]. Previous research has documented a correlative link between metacognition in value-based choice and activity in the frontopolar cortex (FPC)[8,9], but it remains unclear whether this FPC activity indeed causally contributes to choice-related confidence or whether it just relates to confidence without any behavioral implications. To decide whether FPC activity constitutes a functionally relevant neural substrate of metacognition, rather than just a correlate of behaviors associated with metacognition, it is necessary to show that modulating FPC excitability changes metacognitive confidence reports. Previous brain stimulation studies on perceptual decision making have yielded the puzzling finding that FPC disruption improves rather than disrupts metacognitive readout of confidence[10,11], challenging theoretical accounts associating FPC activation with better metacognition. Thus, it remains a matter of controversial debate what role FPC activity plays for representing metacognitive judgments and using them to guide behavior.

In particular, it is unclear how exactly prospective decision making —choices that concern future outcomes—relies on metacognitive awareness of preferences as encoded by the FPC. In the current example, knowing about their individual personal tendency to eat chocolate even though this runs counter to long-term health goals may motivate self-aware humans to restrict access to such temptations, e.g., by avoiding having chocolate at home, a phenomenon referred to as precommitment[12]. The demand for precommitment arises from the possibility of preference reversals, i.e., cases where an individual prefers a more beneficial long-term goal at time $t_0$ (e.g., reducing weight) but switches later at time $t_1$ to preferring a short-term temptation (e.g., eating chocolate) over the long-term goal. Similar to metacognition, precommitment has been linked to FPC activity[13,14], consistent with theoretical claims that metacognition may facilitate precommitment[1,15,16]. Indeed, we recently showed that individuals with stronger metacognitive awareness of their impulsiveness are more likely to precommit[17]. Accordingly, we hypothesize that the FPC implements prospective self-control and restricts access to temptations by strengthening metacognitive awareness of the subjective proneness to preference reversals. Empirical support for this hypothesis would suggest that the FPC plays a causal role for agents using metacognitive knowledge of their economic preferences to optimize choice behavior. Thus, our study substantially extends previous findings about the neural correlates of metacognition[9] by investigating how these neural processes underlying metacognition guide prospective decision making.

To test our general hypothesis with brain stimulation methods, we relied on previous reports that metacognition is associated with frontopolar oscillatory activity in the theta-band[18]. Theta oscillations in dorsolateral (DLPFC) and ventromedial prefrontal cortex (VMPFC) have been linked to representations of choice difficulty or confidence[19–22], consistent with the conjecture that synchronous activity may allow FPC to readout decision-related information from these regions. We thus applied a transcranial alternating current stimulation (tACS) protocol designed to enhance theta-band oscillations in the FPC and tested whether this stimulation (compared to neural-ineffective control stimulation) indeed improves metacognitive judgments.

Metacognition is important in both retrospective and prospective judgements[7]. In retrospective judgments, metacognition is quantified as the accuracy of reported confidence in a choice made. Prospective judgments, in contrast, occur (often implicitly) prior to a decision or performance of a task, and metacognitive accuracy in prospective judgments indicates whether an individual can reliably forecast future decisions or task performance. If FPC theta oscillations constitute a neural substrate for metacognition, theta tACS over FPC should improve metacognition in both retrospective judgments and prospective decision making. We, therefore, assessed the impact of FPC tACS on (1) a confidence accuracy task measuring metacognition as the accuracy of retrospective confidence judgments and (2) a task measuring metacognition as the capacity to assess the risk of preference reversal in prospective self-control decisions about whether to restrict one's access to future temptations. Lastly, because prospective self-control represents only an implicit indicator of metacognition, we also tested the hypothesis that higher metacognitive skills quantified by explicit retrospective confidence judgments predict better prospective self-control, as posited by formal models of precommitment[12,16].

Here, we show that enhancing FPC theta-band oscillations improves both metacognitive accuracies in retrospective judgments concerning intertemporal decision making and the willingness to restrict the access to temptations when anticipating preference reversals. This deepens our understanding of the neural basis of metacognition and suggests a neural link between metacognition and prospective decision-making.

## Results

Thirty-seven participants received theta (5 Hz), control-gamma (80 Hz), and sham tACS over FPC (within-subject design) while performing a confidence accuracy task and a precommitment task. tACS non-invasively modulates brain rhythms in a frequency-specific manner, which allows establishing causal relationships between brain oscillations and behavior[23]. The confidence accuracy task required participants to make choices between a smaller-sooner (SS) and a larger-later (LL) reward (e.g., 8 Swiss francs today versus 10 Swiss francs in 20 days). After each choice, participants indicated their confidence in having made the best decision given their own preferences (Fig. 1A). The precommitment task also required decisions between SS and LL rewards (Fig. 1B), but now participants had to decide between a binding choice of the LL reward (e.g., 10 Swiss francs in 68 days, precommitment) or postponing the decision between the SS and LL reward (e.g., 7 Swiss francs in 28 days or 10 Swiss francs in 68 days). If they chose to postpone the decision, participants were re-contacted after the interval associated with the SS reward (28 days in the example) and had to make the final decision between the two options, with the delays adjusted for the elapsed time (in the example, 7 Swiss francs today versus 10 Swiss francs in 40 days). Thus, individuals who are aware that they may reverse their preference from the LL reward at the first choice to the SS reward in 28 days should prefer to precommit to the LL reward at the first choice opportunity. Note that anticipation of possible preference reversals requires metacognitive access to individual time preferences, such that participants with better metacognitive ability should choose the precommitment option more frequently when the risk of preference reversals increases.

**Theta tACS increases metacognitive sensitivity.** Metacognitive awareness of economic preferences can be determined by

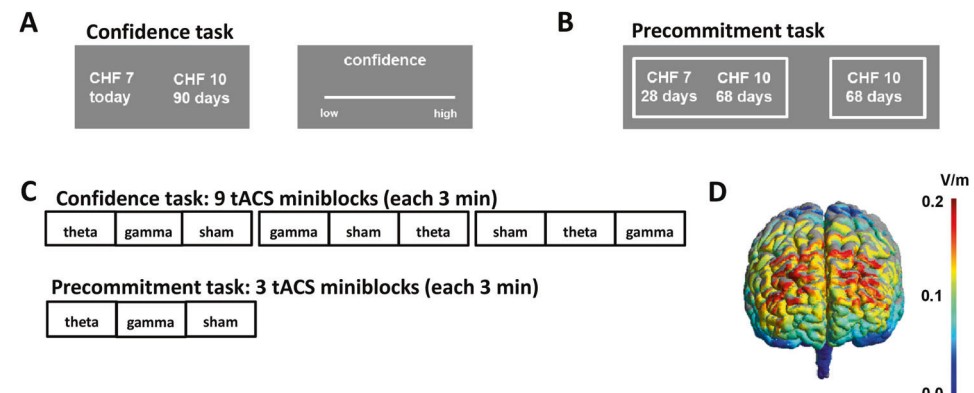

**Fig. 1 Task design and experimental procedure. A** In the confidence accuracy task, participants made binary choices between smaller-sooner (SS; e.g., 7 Swiss francs (CHF) today) and larger-later (LL; e.g., 10 CHF in 90 days) monetary rewards. After each choice, they indicated their confidence in having made the best decision on a scale from 1 to 20. **B** In the precommitment task, participants could either make a binding choice for a LL reward or postpone their decision between a SS and LL reward. In the latter case, participants had to make a final choice after 28 days. The binding choice option should be preferred particularly when participants believe that they might reverse their current preference for the LL reward to the SS reward when having to make their final decision. **C** Participants performed nine blocks of the confidence accuracy task and three mini-blocks of the precommitment task. Except for sham blocks where the current was turned off after 15 s, tACS within each block lasted 160 s (in addition to ramp-up and ramp-down phases of 15 and 5 s, respectively). Blocks were separated by stimulation-free breaks of 30 s. Task order, as well as the order of tACS conditions, were counterbalanced across participants. **D** During the performance of the confidence accuracy and precommitment tasks, participants received theta, gamma, or sham tACS over the frontopolar cortex. The electric field density (in volts per meter, V/m) for our setup was estimated using the Simnibs 2.1 toolbox[39], with warmer colors indicating higher electric field density.

assessing an individual's ability to reliably perceive uncertainty in the decision process. This uncertainty decreases with the difference in subjective value between the choice options, as expressed in the choices of the individual[8]. Specifically, individuals with high metacognitive sensitivity report high levels of confidence when the decision uncertainty they reveal in their choices is low, and low confidence when revealed decision uncertainty is high. In contrast, agents with low metacognitive skills show no systematic relationship between confidence reports and revealed decision uncertainty.

To test our hypotheses, we first determined the subjective value of each choice option by fitting hyperbolic discount functions to the individual choice data, separately for each tACS condition (see "Methods"). Discount parameters $k$ did not differ significantly between stimulation conditions, Wilcoxon rank-sum tests, all $W < 651$, all $p > 0.72$. Moreover, tACS did not affect mean confidence ratings, paired-samples $t$ tests, all $t < 1$, all $p > 0.67$. Thus, there was no evidence that frontopolar stimulation changed time preferences or decision confidence per se. To assess whether tACS affects the relationship between confidence and decision uncertainty, we used a mixed generalized linear model (MGLM). Specifically, we regressed binary choices of LL vs. SS rewards in the confidence accuracy task on predictors for tACS (tACS$_{theta-sham}$ and tACS$_{theta-gamma}$), the signed difference in value (DV, with the subjective value of the LL reward given by the individual hyperbolic discount functions) between LL and SS rewards, confidence ratings, as well as the interactions between these predictors. We assessed how theta stimulation changes behavior relative to both sham tACS (tACS$_{theta-sham}$) and gamma tACS (tACS$_{theta-gamma}$) as an active control condition. The degree to which DV predicts choice is a measure of revealed decision uncertainty (with steeper logistic curves corresponding to lower decision uncertainty). Thus, the strength of the interaction between DV and confidence indicates metacognitive ability, i.e., how reliably an individual can track and report noise in the decision process[8]. Overall, participants showed a significant DV × confidence interaction, beta = 2.22, CI$_{bootstrap}$ = [1.19–3.43], suggesting that they indeed had metacognitive access to their decision uncertainty while making intertemporal choices.

In line with our hypothesis, the ability to report decision uncertainty was significantly increased under theta tACS, both relative to sham tACS, tACS$_{theta-sham}$ × DV × confidence, beta = 2.57, CI$_{bootstrap}$ = [1.32–4.34], and gamma tACS, tACS$_{theta-gamma}$ × DV × Confidence, beta = 1.39, CI$_{bootstrap}$ = [0.37–2.80] (Fig. 2 and Table 1). We note that this result pattern was robust to controlling for potential tACS effects on confidence (which were non-significant, as shown above) by subtracting the mean confidence rating in each tACS condition from confidence scores (tACS$_{theta-sham}$ × DV × Confidence, beta = 2.38, CI$_{bootstrap}$ = [1.44–4.06]; tACS$_{theta-gamma}$ × DV × Confidence, beta = 1.36, CI$_{bootstrap}$ = [0.38–2.81]). An additional MGLM assessing specifically differences between gamma and sham tACS showed a significant tACS$_{gamma-sham}$ × DV × Confidence interaction, beta = 1.14, CI$_{bootstrap}$ = [0.31–2.24]. This may point to some frequency-unspecific stimulation effects. Importantly, however, the effects of theta tACS were significantly stronger than those of gamma tACS, demonstrating specific effects of theta tACS. Individual coefficients (Fig. 2D) suggest that variation in metacognitive sensitivity was larger under theta compared with sham and gamma tACS. This pattern might result from variation in the degree of alignment between individual frontopolar theta rhythms and the applied stimulation frequency of 5 Hz, or from individual differences in the general susceptibility to brain stimulation[24], factors which add to the variation in baseline metacognitive sensitivity. This suggests interesting hypotheses for future experiments that may directly manipulate these factors. Irrespective of these considerations, our present results show that stimulation designed to enhance frontopolar theta oscillations indeed improves the ability to track objective decision uncertainty, supporting the view that these oscillations constitute a causal neural mechanism enabling metacognition of value-based choice processes.

**Theta tACS increases sensitivity to benefits of commitment.** Having established that frontopolar theta tACS improves metacognition, we next asked whether participants apply metacognitive knowledge to restrict their access to temptations in the precommitment task. Metacognitively sophisticated individuals should

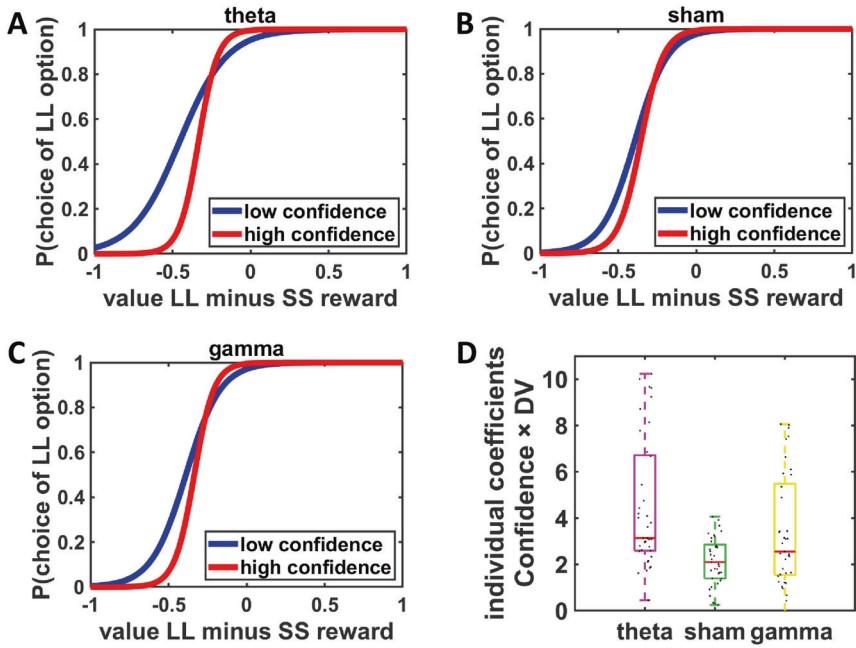

**Fig. 2 Stimulation effects on metacognition based on the results of the mixed generalized linear model for the confidence accuracy task. A** Theta tACS significantly improved metacognitive sensitivity in the confidence accuracy task compared with **B** sham and **C** gamma tACS, indicated by a larger difference between the slopes of logistic curves (these slopes capture revealed decision uncertainty) from low and high confidence decisions. For illustration, we split data into low and high confidence decisions. **D** Individual regression coefficients for the Difference in value (DV) × Confidence interaction, separately for theta, sham, and gamma tACS. DV is defined as the difference between the subjective values of larger-later (LL) and smaller-sooner (SS) rewards. More positive coefficients indicate higher metacognitive accuracy. Colored boxes indicate median and interquartile range, whiskers are defined as 1.5 times the interquartile range or up to the most extreme observed data point falling within this range. Black dots indicate individual data points ($N = 37$ participants).

**Table 1 Results of MGLM explaining choices (1 = LL reward, 0 = SS reward) in the confidence accuracy task.**

| Regressor | Beta | 95% CI$_{bootstrap}$ |
|---|---|---|
| Intercept | 4.27 | [3.30 to 5.84] |
| DV | 11.55 | [8.69 to 16.22] |
| Confidence | 0.64 | [0.19 to 1.11] |
| tACS$_{theta-sham}$ | −0.07 | [−0.58 to 0.45] |
| tACS$_{theta-gamma}$ | 0.15 | [−0.58 to 0.77] |
| DV × Confidence | 2.22 | [1.19 to 3.43] |
| DV × tACS$_{theta-sham}$ | 0.22 | [−1.08 to 1.40] |
| DV × tACS$_{theta-gamma}$ | 0.78 | [−1.02 to 2.71] |
| Confidence × tACS$_{theta-sham}$ | 0.68 | [0.12 to 1.35] |
| Confidence × tACS$_{theta-gamma}$ | 0.34 | [−0.11 to 0.93] |
| DV × Confidence × tACS$_{theta-sham}$ | 2.57 | [1.32 to 4.34] |
| DV × Confidence × tACS$_{theta-gamma}$ | 1.39 | [0.37 to 2.80] |

increasingly prefer to precommit as the risk of preference reversals increases, i.e., the more they switch to preferring the SS over the LL reward at the final choice even though they preferred the LL over the SS option initially. To assess how tACS changes this willingness to precommit with increasing risk of preference reversals, we regressed binary choices in the precommitment task on predictors for tACS (tACS$_{theta-sham}$ and tACS$_{theta-gamma}$), the signed value difference between the LL and SS reward at the first choice instance (DV$_{initial}$), Preference-reversal risk (DV$_{initial}$ − DV$_{final}$), and the interaction terms. DV$_{initial}$ and DV$_{final}$ were again computed based on the individual hyperbolic discount functions. The variable Reversal risk captured the increase of the subjective value of the SS relative to the LL reward at the final choice in case of choice postponement (i.e., for the second choice; DV$_{final}$) compared with the current (initial) choice (DV$_{initial}$), thus measuring the risk of preference reversals for each

combination of reward options (Reversal risk for a given option: DV$_{final}$ − DV$_{initial}$). Again in line with our hypotheses, theta tACS increased the sensitivity of precommitment choices to potential preference reversals relative to sham, tACS$_{theta-sham}$ × Reversal risk, beta = 2.05, CI$_{bootstrap}$ = [0.04–6.24] (Fig. 3A, B and Table 2); this effect was stronger when participants currently preferred the LL over SS reward and might thus reverse their preference from the LL to the SS reward at the time of the final choice, tACS$_{theta-sham}$ × DV$_{initial}$ × Reversal risk, beta = 2.02, CI$_{bootstrap}$ = [0.45–4.20]. Also relative to gamma control stimulation, theta tACS increased sensitivity to preference reversal particularly when the LL option was currently preferred over the SS option, tACS$_{theta-gamma}$ × DV$_{initial}$ × Reversal risk, beta = 2.79, CI$_{bootstrap}$ = [0.15–6.48], even though the lower-level tACS$_{theta-gamma}$ × Reversal risk interaction was not significant, beta = −0.08, CI$_{bootstrap}$ = [−2.40 to 2.82]. A further analysis testing for differences between gamma and sham tACS provided no evidence for potential influences of gamma tACS on precommitment (all confidence intervals contained zero). Taken together, our findings show that FPC theta oscillations motivate precommitment decisions in particular during high risk of preference reversals.

Lastly, we tested whether metacognitive awareness of preference-based choice uncertainty (as measured in the confidence accuracy task) predicts the willingness to make binding LL reward choices in the precommitment task. Formal models of prospective decision-making propose that metacognitive awareness of one's economic preferences should motivate voluntary self-restriction when the risk of preference reversals is high[12,16]. Consistent with the predictions of theoretical accounts, individual estimates of metacognitive sensitivity (individual coefficients for DV × Confidence interaction under sham) significantly correlated with the propensity to precommit with increasing reversal risk (individual coefficients for Reversal risk), $r = 0.61$, $p < 0.001$ (Fig. 3C). We also tested whether low decision

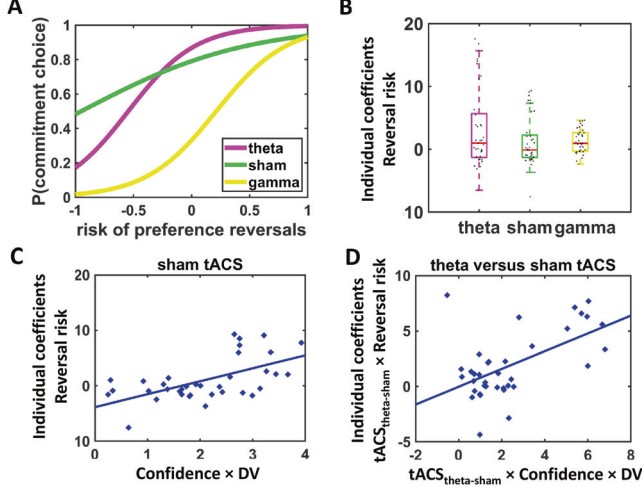

**Fig. 3 Stimulation effects on precommitment (based on the results of the mixed generalized linear model for the precommitment task) and relation to individual differences in impulsiveness. A** Theta tACS enhanced the willingness to precommit with increasing risk of preference reversals in the precommitment task. **B** Individual regression coefficients for the impact of Reversal risk, separately for each tACS condition. Higher coefficients indicate a stronger willingness to precommit with an increasing risk of preference reversals. Colored boxes indicate median and interquartile range, whiskers are defined as 1.5 times the interquartile range or up to the most extreme observed data point falling within this range. Black dots indicate individual data points ($N = 37$ participants). **C, D** Relationship between metacognitive sensitivity (Difference in value (DV) × Confidence) and sensitivity to precommitment demand (Reversal risk). **C** In line with theoretical models, agents with higher metacognitive sensitivity showed the strongest sensitivity to precommitment demands under sham. **D** Improved metacognitive sensitivity under theta relative to sham tACS predicts stronger effects of frontopolar theta tACS on the sensitivity to the risk of preference reversals.

**Table 2 Results of MGLM explaining choices (1 = precommit, 0 = postpone decision) in the precommitment task.**

| Regressor | Beta | 95% CI$_{bootstrap}$ |
|---|---|---|
| Intercept | 1.33 | [−1.02 to 3.98] |
| DV$_{initial}$ | 3.06 | [1.85 to 4.66] |
| Reversal risk | 1.40 | [−0.70 to 3.61] |
| tACS$_{theta-sham}$ | 0.55 | [−0.28 to 2.86] |
| tACS$_{theta-gamma}$ | −2.03 | [−4.12 to 0.52] |
| DV$_{initial}$ × Reversal risk | −1.84 | [−3.76 to 0.12] |
| DV$_{initial}$ × tACS$_{theta-sham}$ | 0.74 | [−0.22 to 2.90] |
| DV$_{initial}$ × tACS$_{theta-gamma}$ | 0.09 | [−1.27 to 2.19] |
| Reversal risk × tACS$_{theta-sham}$ | 2.05 | [0.04 to 6.24] |
| Reversal risk × tACS$_{theta-gamma}$ | −0.08 | [−2.40 to 2.82] |
| DV$_{initial}$ × Reversal risk × tACS$_{theta-sham}$ | 2.02 | [0.45 to 4.20] |
| DV$_{initial}$ × Reversal risk × tACS$_{theta-gamma}$ | 2.79 | [0.15 to 6.48] |

confidence (indicated by participants' mean confidence ratings) predicts higher willingness to precommit, as low confidence might be associated with a high general likelihood of changing one's mind. Contrary to this proposal, there was a positive instead of negative correlation between confidence and precommitment, $r = 0.33$, $p = 0.05$, suggesting that individuals with low choice confidence avoided precommitment to LL rewards and preferred to postpone decisions instead. Lastly, also the strength of tACS effects on metacognition (tACS$_{theta-sham}$ × DV × Confidence)

significantly predicted the impact of tACS on precommitment (tACS$_{theta-sham}$ × Reversal risk), $r = 0.56$, $p < 0.001$ (Fig. 3D). Taken together, these findings support our hypothesis that agents with better metacognitive awareness of their time preferences show increased willingness to restrict their access to temptations to avoid preference reversals.

## Discussion

For consistent (i.e., rational in the economic sense) decision-making, humans need to reliably represent their preferences. Here, we examined the causal neural mechanisms that implement metacognitive awareness of economic preferences and their impact on behavior. First, we show that stimulation designed to enhance frontopolar theta oscillation increases metacognitive sensitivity in economic decision making, providing a causal link between FPC oscillations and metacognition. FPC appears causally involved in reporting subjective confidence during value-based choice, instead of passively representing confidence signals without an active role in metacognitive reporting, informing previous correlational findings[8,9,18]. Because FPC stimulation enhanced the metacognitive link between confidence and preference strength rather than affecting confidence in isolation, theta oscillations in the FPC may process a readout of confidence signals from brain regions that encode decision-related information such as VMPFC[5,25]. Gamma tACS also increased metacognition relative to the sham condition but improvements in metacognition were significantly stronger for theta than for gamma stimulation. Gamma oscillations have previously been found to be nested in theta oscillation in several brain regions[26], and such cross-frequency couplings might be functionally relevant for transferring information across spatial and temporal scales[27]. This notion further supports our interpretation that theta rhythms might implement metacognition via the integration of information from distributed brain networks. Consistent with the idea of cross-frequency couplings, however, gamma oscillations too might be involved in metacognition, though on a different (more local) spatial scale and significantly weaker than theta oscillations. Taken together, our study provides evidence that frontopolar theta oscillations constitute a neural substrate for metacognition, informing the debate on whether metacognition can be considered as a natural kind at the neural level.

Our findings also demonstrate that metacognitive awareness of economic preferences is associated with decision-making. Consistent with the hypothesized role of metacognition for precommitment[15,16] and previous findings[17], our data indicate that decision-makers with higher metacognitive awareness of their temporal preference are increasingly willing to protect themselves from the risk of preference reversals by precommitment. Consistent with the finding that frontopolar theta entrainment improves metacognitive sensitivity, theta oscillations facilitated precommitment decisions as well. Thus, we extend previous findings of FPC involvement in precommitment[13,14] and establish a direct link between the neural implementation of metacognition and voluntarily restricting exposure to temptations. While our findings support the view that metacognition moderates precommitment decisions, we acknowledge that precommitting to the LL reward might not be the optimal choice for all individuals, e.g., for individuals in difficult financial situations. Although individual differences in the financial situation are plausible to have increased variance in participants' behavior, we note that across all participants metacognitive sensitivity predicts higher willingness to precommit to LL options.

Our findings have important implications for neural models of metacognition. FPC was speculated to process a readout of neural signals reflecting confidence and use these signals for communication or behavior control, rather than being involved in the representation or formation of confidence as such[9,11].

Our findings support this view, which is compatible with FPC's proposed position at the top of a hierarchy of control processes[28]. It may receive a readout of confidence signals from VMPFC or DLPFC[8,11,25]. Indeed, theta oscillations have been related to decision-making processes in DLPFC and VMPFC[19,21,29], and prefrontal theta was shown to reflect both objective choice difficulty[20] and self-reported confidence[22]. Theta oscillations might thus enable the FPC to synchronize with these regions and metacognitively access decision-related information like choice difficulty encoded in the theta frequency. It is worth noting that the FPC was related to metacognition in both value-based and perceptual decision making[9,30], supporting a domain-general role in mediating between confidence representations and action planning. Our findings support domain generality by linking FPC to metacognition in both retrospective and prospective value judgments.

We note that besides metacognition the FPC has also been related to other functions that might promote precommitment and future-oriented behavior, for example representing predictive cognitive maps[31–33] or integrating interoceptive signals into goal-directed behavior[34]. The FPC's precise role in cognition is still controversially debated, and it is currently unclear how different views can be integrated. In the context of the current study, however, it seems most parsimonious to explain the stimulation effects on precommitment via the FPC's role for metacognition, given the FPC's well-established role for metacognition in the literature[9] as well as the importance of metacognition for precommitment[17]. We also note that due to the relatively low spatial specificity of tACS, we cannot rule out that the observed results reflect stimulation effects on brain regions adjacent to FPC, including the dorsomedial prefrontal cortex, VMPFC, and DLPFC. However, our current findings provide no evidence for tACS effects on confidence ratings or the degree of hyperbolic discounting, aspects of behavior that have been linked to VMPFC and DLPFC[8,11,35]. Thus, stimulation effects on FPC provide a more parsimonious account of our data, but it remains to be seen whether the stimulation effects concern local activity or coherence of FPC communication with other brain regions. Finally, it is worth mentioning that our experimental design did not include an active control region, which further underlines that we cannot draw strong conclusions regarding the local specificity of the observed stimulation effects.

While our data reinforce theoretical assumptions on the FPC's role for metacognition, they may appear in conflict with previous stimulation studies reporting that disrupting FPC functioning with transcranial magnetic stimulation (TMS) improved, rather than impaired, metacognition in perceptual decision making[10,11]. However, TMS over FPC is commonly perceived as rather aversive by participants, and these studies did not statistically control for stronger discomfort in the FPC compared with control TMS conditions. In contrast, in our study, we employed a less aversive stimulation technique, used active control stimulation over FPC, statistically controlled for tACS-induced discomfort (see "Materials and methods"), and manipulated selectively FPC theta oscillations instead of unspecifically disrupting FPC activation. By showing that frontopolar theta entrainment improves, instead of impairs, metacognitive confidence reports, our results challenge these previous reports and provide evidence for the hypothesized role of FPC functioning in metacognition.

Deficits in metacognitive processing add to the symptoms of several psychiatric disorders, including schizophrenia, addiction, and obsessive–compulsive behavior[2–4]. In obsessive–compulsive behavior, for example, metacognitive deficits appear to exacerbate the dysfunction by impairing behavioral adaptations in response to internal confidence signals[4]. By establishing a causal link between brain oscillations and metacognitive sensitivity, our findings provide a potential mechanistic explanation for the deficits in metacognition and prospective decision-making in these disorders. Thus, impairments in frontal theta oscillations in these disorders might reflect not only deficits in cognitive functioning but also in metacognition[36–38]. Moreover, the link to precommitment may open avenues to pathological impulsiveness.

## Methods

**Participants.** A total of 38 volunteers ($M_{age} = 22.9$ years, range = 19–31, 17 female) participated in this within-subject study. The sample size was determined with a power analysis (alpha = 5%, two-tailed, power = 80%) based on our previous tDCS study on precommitment[14]. One participant terminated prematurely due to tACS-induced side effects (dizziness). The data of this participant were therefore discarded from the analyses. All volunteers gave informed written consent before participating. They received a fixed compensation of 70 Swiss francs plus a performance-dependent bonus (see below). The study was approved by the ethics committee of Canton Zurich.

**Stimuli and task design**

*Confidence accuracy task.* To determine metacognitive awareness of their economic preferences, participants performed a monetary intertemporal choice task (programmed in Matlab using the Cogent toolbox). In each trial, they chose between an immediately available SS reward (0–10 Swiss francs today, in steps of 1 Swiss franc) and a LL reward that was fixed to 10 Swiss francs and was delivered after a delay of 1–180 days (using the following delays: 1, 10, 20, 40, 80, 120, and 180 days). The SS and LL reward options were randomly presented on the left or right side of the screen. Participants chose the left or right option by pressing the left or right arrow key on a standard keyboard within 4 s. Following each choice, participants indicated their confidence that they made the best possible choice on a rating scale from 0 (low confidence) to 20 (high confidence) within 3 s (Fig. 1A). The next trial started after 0.5 s.

*Precommitment task.* As for the confidence accuracy task, participants made choices between SS and LL rewards. One option consisted of a fixed monetary reward of 10 Swiss francs that was delivered after delays of 29–208 days (precommitment option; e.g., "10 Swiss francs in 68 days"; Fig. 1B). When choosing this option, participants received 10 Swiss francs after the indicated delay without having the possibility to reverse their choice. The other option (postpone option) entailed an SS reward of 0–10 Swiss francs delivered after 28 days and a LL reward that was identical to the precommitment option (e.g., "7 Swiss francs in 28 days" or "10 Swiss francs in 68 days"). If participants chose this option, they were re-contacted by the experimenter via email after 28 days and were asked to make a choice between the SS and LL rewards, with the delays adjusted for the 28 days that had passed (in the current example, "7 Swiss francs today" or "10 Swiss francs in 40 days"). If they selected the precommitment option, they received information about the chosen option via email after 28 days without the possibility to reverse their choice. The postpone option thus allowed participants to reverse their preference after 28 days, whereas with the precommitment option they made a binding choice for the LL reward.

**Procedure.** Participants performed the confidence accuracy task and the precommitment task in counterbalanced order (Fig. 1C). The confidence accuracy task included 180 trials, with each combination of SS and LL options repeated three times (once in each tACS condition). Participants performed the task in 9 mini-blocks of 20 trials while undergoing theta, gamma, and sham tACS. The order of stimulation conditions was counterbalanced using Latin square methods. In a similar fashion, the precommitment task entailed 90 trials, divided into mini-blocks of 30 trials each.

Each mini block started with a ramp-up period of 15 s for the tACS current. Totally, 30 s after the start of the stimulation (except for the sham condition where the current was ramped down after the ramp-up phase), participants performed the confidence accuracy or precommitment task for 125 s. In addition to the tasks, participants also had to indicate the perceived aversiveness of the stimulation after each mini-block on a rating scale from 0 (not aversive at all) to 20 (very aversive). There was no evidence that aversiveness ratings differed between tACS conditions, all $t < 1$, all $p > 0.38$. Totally, 5 s after task performance, the current was ramped down over a period of 5 s. Thus, the total stimulation duration in each mini-block was 180 s, which allowed us to minimize the risk of stimulation-induced physiological after-effects. The next block started after a stimulation-free interval of 30 s.

At the end of the experiment, participants indicated the tDCS-induced discomfort, whether they perceived flickering during tACS, as well as whether the discomfort or flickering affected their task performance on Likert scales from 1 to 9. The mean ratings reported for discomfort and flickering were 4.1 and 5.6, respectively, but participants reported only low to moderate disturbing influences of discomfort (mean = 3.4) and flickering (mean = 2.9) on task performance. In order to control for any influences of tACS-induced irritations on task performance, we added the individual ratings for the impact of discomfort and

flickering on task performance as control variables to all statistical models. For the payment, one trial of the two tasks was randomly selected and the corresponding amount was paid out after the associated delay. In case a trial of the precommitment task was chosen where participants had decided to postpone the decision, they were re-contacted 28 days after the experimental session and had to make a final choice between the SS and LL option. If participants chose an option to be paid out after the experiment (either in the confidence accuracy or in the precommitment task), the given amount was sent to the participant via mail.

**tACS protocol.** We applied tACS using an 8-channel tDCS stimulator (DC-stimulator MC, neuroConn, Ilmenau, Germany). As previous studies suggest both metacognition and precommitment to be implemented by FPC[9,14,18], we placed a smaller active $5 \times 7$ cm$^2$ electrode over electrode position Fpz and a larger $10 \times 10$ cm$^2$ reference electrode over CPz according to the international 10–20 system (Fig. 1D). Current modeling using the Simnibs 2.1 toolbox[39] suggests that with this electrode setup current density is strongest in FPC while stimulation effects under the reference electrode are negligible. The electrodes were fixed to the participants' heads by rubber straps. We used larger reference than active electrodes to minimize the stimulation effect at the vertex relative to the FPC site. We stimulated participants in the theta band (5 Hz) and gamma band (80 Hz) frequency with a current strength of 2 mA (peak-to-peak). The control frequency of 80 Hz was determined in pilot experiments to match the tACS-induced discomfort and phosphenes between theta and control tACS. We note that phosphenes appear to affect performance mainly in visual perception tasks[40,41], and it seems much less likely that phosphenes would affect (and in fact improve) value-based decision making, but we cannot logically rule out this possibility.

**Data analysis.** Data were analyzed with mixed generalized linear models (MGLMs) implemented in R using the lme4 package. The advantage of MGLMs over other statistical procedures is that MGLMs provide a better account of the full variation in choice data sets with binomially distributed binary dependent variables, compared to participant-specific aggregated approaches that neglect intra-individual variability on the trial level. In MGLMs statistical inferences are based on group-level fixed effect estimates while accounting for inter-individual variation via random effects. We assessed the significance of fixed effects by the 95% confidence intervals (CI$_{bootstrap}$) determined via parametric bootstrap (implemented by the bootMer function in R), which provides more reliable results than $p$ values based on Wald statistics. Importantly, our hypothesis that theta tACS improves metacognition is only supported by significant results for the comparisons between theta tACS and *both* sham tACS (passive control) and gamma tACS (as active control), and not by significant results for just one of these comparisons. Alpha correction for multiple comparisons was therefore not required[42].

In the confidence accuracy task, we measured participants' metacognitive awareness of their decision noise (uncertainty) following a previously described approach[8]. For that purpose, we first estimated each individual's time preferences by fitting a hyperbolic discount function to the choices in the confidence accuracy task, separately for each tACS condition (Eq. 1):

$$SV_{LL} = \frac{LL \text{ reward magnitude}}{1 + k \times \text{delay}}, \tag{1}$$

where $SV_{LL}$ is the discounted subjective value of the LL reward and k is a participant-specific constant that indicates the steepness of the discount function (discount factor). To translate subjective value into choices, we fitted a standard softmax function to each participant's choices:

$$P(\text{choice of LL option}) = \frac{1}{1 + e^{-\beta_{temp} \times (SV_{LL} - SV_{SS})}} \tag{2}$$

this function captures the likelihood of choosing the LL reward option as a function of the difference between the subjective value of the LL reward option ($SV_{LL}$) and the SS reward option ($SV_{SS}$), with the inverse temperature parameter $\beta_{temp}$ capturing the slope of the function, i.e., how strongly participants relied on this value difference for their choices. Individual parameters were estimated with a Bayesian approach (4 chains with 10,000 samples, the first 2000 samples were used as burn-in) using the hBayesDM package[43]. All chains converged, as indicated by $\hat{R}$ values below 1.01. Moreover, discount parameters in the three tACS conditions were strongly correlated with each other, all $r > 0.95$, all $p < 0.001$, providing evidence for the reliability of individual parameter estimates. To measure metacognitive access to individual time preferences, we computed the difference between the value of the SS reward and the subjective value of the LL reward ($SV_{LL} - SV_{SS}$) based on the individual discount factors. We then performed an MGLM regressing binary choices (1 = LL reward, 0 = SS reward) on fixed-effects predictors for tACS, subjective value difference (DV = $SV_{LL} - SV_{SS}$), confidence ratings, as well as all interactions. We analyzed tACS effects with two separate predictors, measuring the impact of theta relative to sham (tACS$_{theta-sham}$) as well as theta relative to gamma stimulation (tACS$_{theta-gamma}$). Confidence was z-standardized on the subject level to control for individual differences in metacognitive bias (degree of confidence), as recommended when assessing metacognitive sensitivity[44,45]. The interaction between value difference and confidence indicates the degree to which participants are aware of objective decision uncertainty in the choice process and thus constitutes a measure of metacognitive

accuracy: the stronger the interaction between decision uncertainty and confidence ratings, the more reliably an individual is able to track preference strength[8]. Assessing the impact of tACS on the DV × Confidence interaction effect thus allowed us to test whether stimulation modulated the degree to which participants were metacognitively aware of their time preferences. As predictors of no interest, we also modeled the order in which the confidence accuracy and precommitment tasks were performed, the order of the tACS conditions, as well as the tACS-induced discomfort rated at the end of each block and the perceived impact of discomfort and flickering on task performance (as given by the post-experiment ratings). As random effects, we included participant-specific intercepts as well as random slopes for all fixed-effect predictors varying on the individual level.

In the precommitment task, we asked whether tACS changed the sensitivity to the expected benefit from precommitment. We conducted an MGLM that regressed choices in the precommitment task (0 = postpone option, 1 = precommitment option) on predictors for tACS, Reversal risk, the value difference between SS and LL reward (DV$_{initial}$), and all interaction terms. We computed the reversal risk on each trial by subtracting the value difference between SS and LL reward in the current perspective (DV$_{initial}$) from the value difference between these options in the perspective of 28 days later (when participants made a definite choice between the options; DV$_{final}$) based on the individually determined discount factors in the confidence accuracy task. A higher score indicates a higher risk of preference reversals (i.e., that a participant prefers the LL reward in the experimental session and the SS reward in 28 days) and thus a higher expected benefit from precommitting to the LL reward. Again, we also added predictors for task order, tACS-induced discomfort, the impact of flickering, and perceived aversiveness on task performance, as well as random slopes for all predictors varying on an individual level in addition to participant-specific random intercepts. Thus, the MGLM for the precommitment task included the same predictors as the MGLM for the confidence accuracy task, except that we replaced the predictor for confidence ratings (which were not measured in the precommitment) with a predictor for reversal risk, as the goal of the precommitment task was to measure the willingness to precommit as a function of the risk of preference reversals. If theta tACS increases the sensitivity to potential preference reversals, this should be expressed by significant tACS effects on the Reversal risk × DV$_{initial}$ interaction. Finally, we assessed potential tACS effects on aversiveness ratings at the end of each block with an MGLM that regressed these ratings on fixed-effects predictors for tACS, modeling also random slopes for tACS in addition to random intercepts.

**Reporting summary.** Further information on experimental design is available in the Nature Research Reporting Summary linked to this paper.

## Data availability
The data that support the findings of this study are available on Open Science Framework (https://osf.io/m57yg/)[46]. Source data are provided with this paper.

## Code availability
Code to reproduce the findings of this study is available on Open Science Framework (https://osf.io/m57yg/)[46]. Source data are provided with this paper.

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

## Acknowledgements
P.N.T. received funding from the Swiss National Science Foundation (Grants 100019_176016, 100014_165884, and CRSII5_177277) and from the Velux Foundation (Grant 981). C.C.R. received grant support from the Swiss National Science Foundation (100019L_165973) and from the European Research Council (ERC) under the European Union's Horizon 2020 research and innovation program (grant agreement No. 725355; ERC Consolidator Grant BRAINCODES). A.S. received an Emmy Noether fellowship (SO 1636/2-1) from the German Research Foundation and a grant from the Richard Büchner Foundation.

## Author contributions
A.S., M.M., C.C.R. and P.N.T. designed the research; A.S. and M.M. performed the research; A.S. analyzed the data; A.S., M.M., C.C.R. and P.N.T. wrote the paper.

## Funding

## Competing interests
The authors declare no competing interests.
