## [Peer Review File · Nature Communications]

REVIEWER COMMENTS

Reviewer #1 (Remarks to the Author):

Soutschek et al. present novel evidence for a link between metacognition of value-based decision-making, and the ability to “precommit” to avoid preference reversals later in time. A link between metacognitive awareness and self-control has long been suspected (also in classical models from behavioural economics), and there have been intriguing hints in empirical data that similar processes might be involved (eg common involvement of frontopolar cortex). By using a clever design in which both metacognition of value-based decision-making and awareness of potential preference reversals (as indexed by precommitment decisions) are measured in the same subjects, the authors provide novel and clear evidence for a link between these phenomena. A causal manipulation of brain activity hypothesised to support metacognition (frontopolar theta oscillations) provides a strong test of their common involvement in intertemporal choice.

In general the within-subject design and analyses were compelling and clearly presented. The novel and elegant behavioural paradigm will no doubt catalyse further investigations of the link between metacognition and precommitment. I have one major comment about the logic of the results on metacognitive sensitivity, and what role bias/confidence level might play here, which should be addressable with further analysis.

On p. 5 the key hypothesis is laid out – that participants who have better metacognitive ability (i.e. increased meta sensitivity) should choose the precommitment option more frequently, because they are more aware of potential future reversals. I wasn't completely sold on this. The metacognitive sensitivity being measured here is about confidence in preferences at time t – not anything about awareness of potential reversals at time $t+1$. In other words, it tells us about the confidence-value alignment when precommitment is not available, rather than when it might be useful in maintaining internal consistency of preferences. I guess an argument could be made (though I could not find this made explicit in the paper) that lower confidence in these initial choices indexes the likelihood of a future change of mind, and that it's these choices which should be associated with precommitment when it becomes available. But shouldn't the prediction then be that lower confidence level / bias predicts the extent of precommitment usage, rather than confidence-value alignment or meta sensitivity?

Following the same line of reasoning, it seems important to assess whether the tACS altered confidence level, not only metacognitive sensitivity. The confidence data were z-scored in the sensitivity analysis which makes sense, but I could not find un-normalised confidence level compared between conditions. Given the above line of reasoning, it would be important to know whether it's metacognitive sensitivity, bias or a combination of both that changes across tACS conditions, and which component(s) predicts precommitment usage.

A final, related point is on the assumed neural basis of the tACS effect. Throughout the paper this is referred to as FPC, and this makes sense based on previous imaging of both metacognition and precommitment. But is it not possible (due to the diffuse nature of tACS) that the effect is due to changes in vmPFC and effects on confidence per se, rather than metacognitive sensitivity and FPC? The discussion on p. 9 claims otherwise (based on reverse inference from a change in metacognitive sensitivity) but this seems a somewhat weak argument. I was also not convinced by the remainder of the discussion suggesting a network-level model of the findings (eg with interactions between vmPFC, FPC and DLPFC) given that these regions are all close to one another, their boundaries are debated (eg some parts of vmPFC might extend into polar area 10, etc) and most importantly, I suspect that the precision of tACS cannot rule out effects on these neighbouring regions (eg Figure 1D shows field density spreading laterally and dorsally, and presumably medially too?). So I suggest toning down these conclusions and including caveats on localisation.

Minor points

- I felt more could be done in the main text to explain how DV was extracted (ie from a model of temporal discounting). It's in the Methods, but would help the reader if it was highlighted earlier.
- Similarly, could some context be provided for the DV terms in the preference-reversal risk calculation on p. 7. As far as I understand it from the Methods, these are both calculated from a single discounting model fitted to the confidence task data, but this would be useful to make explicit.
- I could not find the statistics for the comparison of precommitment sensitivity (ie tACS x reversal risk) between theta and gamma conditions (only between theta and sham).

Reviewer #2 (Remarks to the Author):

This interesting report describes the results from two closely related experiments on metacognition, the FPC, and Theta Rhythms. These derive from recent work (some by this group) in the literature arguing that coherent FPC theta relates to the ability of subjects to report when they are near indifference in the underlying subjective valuations; effectively strengthening the correlation between a subjective value-based model of choice noise and a subjects's reported "decision confidence." The current paper extends that work on decision confidence and relates it to precommitment "accuracy," the willingness of subjects to commit actively to future decisions when they are aware that their own decisional process might reverse in the future. The authors tie these two sets of results together with the psychological construct of meta-cognition, using their results to strengthen their argument that FPC theta is a neurobiological correlate of metacognition.

One of the most challenging (and to me the most interesting) issues engaged by the paper is the issue of whether "metacognition" is a meaningful "kind" at the neurobiological level. Psychologists have for some time argued that metacognition is a coherent conceptual structure that can be used to tie together a group of related findings. Studies of confidence ratings (albeit with somewhat conflicting results) have been used to tie confidence rating "accuracy" to FPC activity levels, and thus to metacognition. But is metacognition a logical "kind" at the neurobiological level? I honestly don't know what I believe, and this paper is to be commended for trying to work on this issue. That having been said, I want to note to the authors that they don't really make it clear that this is a key goal of the paper and so the paper reads more as a pair of loosely related and somewhat disjointed experiments than as an investigation of the neurobiological qualia of metacognition. I think that this makes the paper a bit confusing to someone reading it quickly, and I would suggest that the authors think about that observation in preparing future drafts.

General Comments

In line with my comments in the preceding paragraph, I would urge the authors to consider a reorganization of the presentation into two experiments on metacognition and the FPC. They could start with a clear statement of the metacognitive hypothesis of the FPC and then work through their two experiments. I think this would come across as much less disjointed, and it would be clearer why these two experiments go together. As writing one kind of wonders if the experiments are being presented together simply because they were run together.

Turning to the experimental results themselves, I think that the biggest disappointment is that while the theta stimulation does have a big effect, which is cool, so does gamma stimulation. The authors note appropriately that this is only significantly true in the first experiment, but to my eye it looks like it's probably true in the second experiment, but that it does not rise to significance for reasons of inadequate power. That is really, to my mind, the greatest weakness of the paper. One really wants to show that increasing theta increases all of these metacognitive behaviors and that decreasing theta reduces these behaviors. That is really the result that the authors need to strive for in the future. One cannot help but wonder (and let me say that I am no tACS expert) whether there is a stimulation parameter set that would be better for disrupting theta than gamma? What if the tACS protocol had

plenty of power at theta frequencies but either with coherent or scrambled phase, might that be a better approach? I'm sure the authors know better than I, but in the long run they will have to make this part of the argument more convincing.

More Specific Comments

I thought that the most interesting and entirely unexplored result in the paper is the observation that the theta stimulation has a huge effect on variance. Just looking at figs 2 and 3 I was struck by how different the theta effect was in this regard. This stands out just from seeing how much bigger the difference is between the medians than between the means. I felt that this was a huge missed opportunity. In any revision the authors need a systematic exploration of this feature of the data. It probably has much to do with mismatches between their exact stimulation protocol and the intrinsic neural dynamics of the individual subjects. If some subjects have higher or lower theta, if phase is aligned or misaligned, if location of stimulation and location of the relevant theta are aligned, these things all probably control the efficacy of stimulation and their data suggests that this is true – but there isn't a hint of this in the paper. I think that in any revision that simply has to be addressed.

The comment above really also goes to the question of how best to analyze these data. With a linear regression approach as done here or with a more non-parametric approach. There are so many assumptions in the GLM and we have so little idea what is really going on with the tACS. If the authors want a reader to stick with the GLM they should say a bit about the distributional structure of the data supporting this approach

At a more methodological level I also had one small question: How were the subjects actually paid in experiment 2? Paying out Smaller-sooner vs Larger-later experiments is annoying and important and I just wanted to understand if the same mechanism was used for now, 28d and longer delays, and what was that mechanism.

Reviewer #3 (Remarks to the Author):

In this study the authors use tACS to investigate the relationship of frontopolar theta oscillations and metacognition of economic preference. The authors came up with an interesting design to address this timely topic in the field of human decision-making. The results could be potentially interesting and provide us with additional insight into the mechanisms underlying our ability to control our own decision process. Below you will find my main comments:

Throughout the manuscript the authors claim that the main effects are related to increased theta power in frontopolar cortex. However, the spatial specific claims seem to be unwarranted with the current methods used (i.e., not using HD-tDCS). The strength of tACS might also not be in spatial specificity but in enhancements of coherence between brain areas. In addition, the plot showing the modeling of the electric field density also shows large spread to DLPFC.

Some of the main findings have p-value of $p=0.04$ (precommitment results) and $p=0.049$ (theta gamma difference in the confidence task). It is a bit unclear at the moment how stable these effects are. Maybe a bootstrap procedure could help with this.

The individual time preference itself can also be changed by stimulation. Each individual's time preferences is fitted by a hyperbolic discount function to the choices in the confidence task. However, it is unclear what data has been used for the fit (e.g., all data, or sham data) and whether potential changes in preference due to stimulation occur.

How is the z-transform performed on the confidence scores? If this has been done per participant

across conditions, it could be that specific stimulation protocols affected response bias instead of metacognitive sensitivity (overall higher/lower confidence reports irrespective of DV), see Masson, M. E. J., and Rotello, C. M. (2009).

Recently there are many interesting findings with respect to gamma nested in theta. By using gamma as an active control causes some issues with respect to these observations. In case theta serves communication between brain areas by nesting gamma bursts in each cycle, the present finding of a significant gamma effect might support such a mechanism.

FPC has many other functions that could be relevant in the present design. Some of them: the formation of internal predictions and forecasting, and the "somatic marker" hypothesis (see for instance Momennejad - Current Opinion in Behavioral Sciences, 2020; Wokke & Ro - Journal of Neuroscience, 2019, T Poppa, A Bechara - Current opinion in behavioral sciences, 2018).

The design assumes that the LL is the better option. However, this all depends on the participants. In case they are students knowing that in 28 days it's the end of the month and they need the money, the value parameters change. Therefore to postpone the decision and see how the financial situation is at the moment could in some instances be the better option, depending on circumstances. Alternatively, participants might choose the option "not to choose again". Having nothing to do with the actual value.

What does "this effect was numerically stronger" mean?

Reviewer #4 (Remarks to the Author):

In the manuscript by Soutschek et al. the authors explore the effect of theta tACS on choice behavior. Below I focus on several methodological aspects of the study. Overall, there seems to be only weak support that tACS affected task performance in a meaningful and specific manner.

1. The authors explored two active stimulation conditions (plus sham) during two tasks, both with multiple readouts. The statistical significance of the tACS effect was established using a series of linear models (22 reported regressors between Tables 1 and 2). However, I don't see any attempts to correct for the multiple comparisons that arose from so many statistical models in the paper.

2. The main effect of the theta stimulation itself is not significant in either task ($p=0.94$ in the confidence task; $p=0.08$ in the precommitment task, both not corrected) and only reaches significance in interaction with other statistical factors (choice confidence, subjective value difference, or reversal risk). Importantly, these other factors are not the same between the two tasks, which further complicates the interpretation.

3. Although the authors performed a computational model of the stimulation, this is not well described in the manuscript. I can only see one corresponding visualization (Figure 1, panel D). For some reason, the values on the figure are restricted between 0.1 and 0.2 V/m, which are probably the half-maximum and maximum values. Such partial presentation and lack of any quantitative analysis give little confidence in what brain areas were actually stimulated. In addition, there is no control stimulation montage (location) employed in the study. So, one cannot confidently attribute any stimulation effects to a specific brain area.

4. One major confound that could have impacted the results here are phosphenes (activation of the eye's retina by the alternating currents). The authors show no results of post-stimulation questionnaires regarding the presence of phosphenes, although the stimulation electrodes were right above the eyes.

5. Despite the rather complex nature of the cognitive task, which included a choice and a confidence

rating, the authors opted to use a very limited number of task trials - 30 or 60 trials per stimulation condition in the precommitment or confidence tasks, respectively. That is most likely not enough to reliably estimate individual parameters of metacognitive effects using a Bayesian approach.

Reviewer #1:

Remarks to the Author:

Soutschek et al. present novel evidence for a link between metacognition of value-based decision-making, and the ability to “precommit” to avoid preference reversals later in time. A link between metacognitive awareness and self-control has long been suspected (also in classical models from behavioural economics), and there have been intriguing hints in empirical data that similar processes might be involved (eg common involvement of frontopolar cortex). By using a clever design in which both metacognition of value-based decision-making and awareness of potential preference reversals (as indexed by precommitment decisions) are measured in the same subjects, the authors provide novel and clear evidence for a link between these phenomena. A causal manipulation of brain activity hypothesised to support metacognition (frontopolar theta oscillations) provides a strong test of their common involvement in intertemporal choice.

In general the within-subject design and analyses were compelling and clearly presented. The novel and elegant behavioural paradigm will no doubt catalyse further investigations of the link between metacognition and precommitment. I have one major comment about the logic of the results on metacognitive sensitivity, and what role bias/confidence level might play here, which should be addressable with further analysis.

Response: We thank the reviewer for the positive evaluation of our study and for the insightful comments. We believe that addressing these comments further improved the quality of the manuscript.

On p. 5 the key hypothesis is laid out – that participants who have better metacognitive ability (i.e. increased meta sensitivity) should choose the precommitment option more frequently, because they are more aware of potential future reversals. I wasn't completely sold on this. The metacognitive sensitivity being measured here is about confidence in preferences at time t – not anything about awareness of potential reversals at time $t+1$. In other words, it tells us about the confidence-value alignment when precommitment is not available, rather than when it might be useful in maintaining internal consistency of preferences. I guess an argument could be made (though I could not find this made explicit in the paper) that lower confidence in these initial

choices indexes the likelihood of a future change of mind, and that it's these choices which should be associated with precommitment when it becomes available. But shouldn't the prediction then be that lower confidence level / bias predicts the extent of precommitment usage, rather than confidence-value alignment or meta sensitivity?

Response: We thank the reviewer for this insightful comment. We agree that participants with better metacognitive abilities should not precommit more frequently overall, but primarily when they anticipate preference reversals. Please note that the correlation between metacognition and precommitment reported in the paper does not operationalize precommitment by the overall frequency of binding choices – it rather employs the individual slopes of precommitment choices across the predictor Reversal risk (which indexes the willingness to precommit as a function of the propensity to change one's mind). Our findings thus suggest that individuals with higher metacognitive accuracy do not precommit more often overall but only when preference reversals are likely. In other words, when facing the decision whether or not to precommit at time t , individuals may mentally simulate their preferences at time $t+1$. Individuals with better metacognitive knowledge of their preferences (and thus more accurate simulations of their future preferences) are more likely to anticipate preferences reversals and may therefore decide to precommit. In line with the reviewer's suggestion, we formulate the description of this correlation analysis and the resulting hypothesis more clearly now (see text below).

Moreover, we also tested the reviewer's suggestion that low confidence, rather than metacognitive sensitivity, might predict the willingness to precommit. However, contrary to the reviewer's assumption, mean confidence scores showed a positive, rather than negative, correlation with individual coefficients for the influence of reversal risk on precommitment, $r = 0.33$, $p = 0.05$. Thus, individuals with low decision confidence did not show a stronger preference for precommitment, but instead preferred to postpone the final decision. We note that the correlation between precommitment and confidence ($r = 0.33$) is weaker than the correlation between precommitment and metacognitive sensitivity ($r = 0.61$). In our view, this suggests that not the level of confidence per se, but rather the degree of metacognitive insight determines an individual's willingness to precommit when the risk of preference reversals is high.

To address both of these points, we have made the following changes to the manuscript (p.9-10):

“Formal models of prospective decision making propose that metacognitive awareness of one’s economic preferences should motivate voluntary self-restrictions when the risk of preference reversals is high^{12, 16}. Consistent with the predictions of theoretical accounts, individual estimates of metacognitive sensitivity (individual coefficients for DV × Confidence interaction under sham) significantly correlated with the propensity to precommit with increasing reversal risk (individual coefficients for Reversal risk), $r = 0.61$, $p < 0.001$ (Figure 3C). We also tested whether low decision confidence (indicated by participants’ mean confidence ratings) predicts higher willingness to precommit, as low confidence might be associated with a high general likelihood of changing one’s mind. Contrary to this proposal, there was a positive instead of negative correlation between confidence and precommitment, $r = 0.33$, $p = 0.05$, suggesting that individuals with low choice confidence avoided precommitment to LL rewards and preferred to postpone decisions instead.”

We also formulated our hypothesis more clearly on p.6:

“anticipation of possible preference reversals requires metacognitive access to individual time preferences, such that participants who have better metacognitive ability should choose the precommitment option more frequently when the risk of preference reversals increases.”

Following the same line of reasoning, it seems important to assess whether the tACS altered confidence level, not only metacognitive sensitivity. The confidence data were z-scored in the sensitivity analysis which makes sense, but I could not find un-normalised confidence level compared between conditions. Given the above line of reasoning, it would be important to know whether it’s metacognitive sensitivity, bias or a combination of both that changes across tACS conditions, and which component(s) predicts precommitment usage.

Response: We agree that it is important to check whether frontopolar tACS affected (un-normalized) confidence ratings. In the revised manuscript, we now report additional analyses that provide no evidence that tACS conditions differed with respect to mean confidence. Thus, it seems that frontopolar theta tACS selectively affects metacognitive sensitivity rather than

metacognitive bias (as tACS did not alter mean confidence ratings). Moreover, precommitment is more strongly predicted by metacognitive sensitivity rather than confidence per se (see response to previous comment).

We report these results on p.6-7:

“Moreover, tACS did not affect mean confidence ratings, paired-samples t-tests, all $t < 1$, all $p > 0.67$. Thus, there was no evidence that frontopolar stimulation changed time preferences or decision confidence per se.”

A final, related point is on the assumed neural basis of the tACS effect. Throughout the paper this is referred to as FPC, and this makes sense based on previous imaging of both metacognition and precommitment. But is it not possible (due to the diffuse nature of tACS) that the effect is due to changes in vmPFC and effects on confidence per se, rather than metacognitive sensitivity and FPC? The discussion on p. 9 claims otherwise (based on reverse inference from a change in metacognitive sensitivity) but this seems a somewhat weak argument. I was also not convinced by the remainder of the discussion suggesting a network-level model of the findings (eg with interactions between vmPFC, FPC and DLPFC) given that these regions are all close to one another, their boundaries are debated (eg some parts of vmPFC might extend into polar area 10, etc) and most importantly, I suspect that the precision of tACS cannot rule out effects on these neighbouring regions (eg Figure 1D shows field density spreading laterally and dorsally, and presumably medially too?). So I suggest toning down these conclusions and including caveats on localisation.

Response: We agree that the relatively diffuse nature of tACS does not allow strong claims regarding local specificity. In the discussion section, we accordingly added a caveat clarifying that besides frontopolar cortex (BA 10) our stimulation protocol might have affected also other regions like dorsomedial prefrontal cortex (BA 9), VMPFC, or DLPFC. While parts of the dorsomedial prefrontal cortex may also be involved in metacognitive processing (Vaccaro and Fleming, 2018), VMPFC and DLPFC have previously rather been associated with confidence and self-control, respectively. While the lack of significant stimulation effects on confidence and self-control (indicated by discount parameters k) thus speaks against strong tACS effects on these regions, we cannot completely rule out this possibility (p.12).

“We also note that due the relatively low spatial specificity of tACS, we cannot rule out that the observed results reflect stimulation effects on brain regions adjacent to FPC, including dorsomedial prefrontal cortex, VMPFC, and DLPFC. However, our current findings provide no evidence for tACS effects on confidence ratings or the degree of hyperbolic discounting, aspects of behavior that have been linked to VMPFC and DLPFC^{8, 11, 35}. Thus, stimulation effects on FPC provide a more parsimonious account of our data, but it remains to be seen whether the stimulation effects concern local activity or coherence of FPC communication with other brain regions.”

Minor points

- I felt more could be done in the main text to explain how DV was extracted (ie from a model of temporal discounting). It's in the Methods, but would help the reader if it was highlighted earlier.

Response: We apologize that this was not sufficiently clear in the previous manuscript. We now clarify that discounted subjective values were computed by fitting hyperbolic discount functions to the individual choice data, separately for each tACS condition, in order to control for potential stimulation effects on hyperbolic discounting (p.6):

“To test our hypotheses, we first determined the subjective value of each choice option by fitting hyperbolic discount functions to the individual choice data, separately for each tACS condition (see Materials and Methods). Discount parameters k did not significantly differ between stimulation conditions, Wilcoxon rank sum tests, all $W < 651$, all $p > 0.72$.”

- Similarly, could some context be provided for the DV terms in the preference-reversal risk calculation on p. 7. As far as I understand it from the Methods, these are both calculated from a single discounting model fitted to the confidence task data, but this would be useful to make explicit.

Response: In the revised manuscript, we clarify that the difference in values (DV) was based on the discounted subjective reward values as given by the individual hyperbolic discount functions (p.8):

“ DV_{initial} and DV_{final} were again computed based on the individual hyperbolic discount functions.”

- I could not find the statistics for the comparison of precommitment sensitivity (ie tACS x reversal risk) between theta and gamma conditions (only between theta and sham).

Response: We apologize that in the previous manuscript this interaction effect had been reported only in Table 2, not in the main text. In the revised manuscript, it is now reported in the main text as well (p.9). We note that the non-significant result of the $tACS_{\text{theta-gamma}} \times \text{Reversal risk}$ interaction does not change our conclusions given the significant higher-order $tACS_{\text{theta-gamma}} \times DV_{\text{initial}} \times \text{Reversal risk}$ interaction.

“Also relative to gamma control stimulation, theta tACS increased sensitivity to preference reversal particularly when the LL option was currently preferred over the SS option, $tACS_{\text{theta-gamma}} \times DV_{\text{initial}} \times \text{Reversal risk}$, $\beta = 2.79$, $CI_{\text{bootstrap}} = [0.15; 6.48]$ (even though the lower-level $tACS_{\text{theta-gamma}} \times \text{Reversal risk}$ interaction was not significant, $\beta = -0.08$, $CI_{\text{bootstrap}} = [-2.40; 2.82]$).”

Reviewer #2:

Remarks to the Author:

This interesting report describes the results from two closely related experiments on metacognition, the FPC, and Theta Rhythms. These derive from recent work (some by this group) in the literature arguing that coherent FPC theta relates to the ability of subjects to report when they are near indifference in the underlying subjective valuations; effectively strengthening the correlation between a subjective value-based model of choice noise and a subjects's reported "decision confidence." The current paper extends that work on decision confidence and relates it to precommitment "accuracy," the willingness of subjects to commit actively to future decisions when they are aware that their own decisional process might reverse in the future. The authors tie these two sets of results together with the psychological construct of meta-cognition, using their results to strengthen their argument that FPC theta is a neurobiological correlate of metacognition.

Response: We thank the reviewer for judging our study as interesting. Below we provide detailed responses to each of the reviewer's comments.

One of the most challenging (and to me the most interesting) issues engaged by the paper is the issue of whether "metacognition" is a meaningful "kind" at the neurobiological level. Psychologists have for some time argued that metacognition is a coherent conceptual structure than can be used to tie together a group of related findings. Studies of confidence ratings (albeit with somewhat conflicting results) have been used to tie confidence rating "accuracy" to FPC activity levels, and thus to metacognition. But is metacognition a logical "kind" at the neurobiological level? I honestly don't know what I believe, and this paper is to be commended for trying to work on this issue. That having been said, I want to note to the authors that they don't really make it clear that this is a key goal of the paper and so the paper reads more as a pair of loosely related and somewhat disjointed experiments than as an investigation of the neurobiological qualia of metacognition. I think that this makes the paper a bit confusing to someone reading it quickly, and I would suggest that the authors think about that observation in preparing future drafts.

Response: We thank the reviewer for pointing to this important theoretical issue. We agree that it is not straightforward to define metacognition and its links to confidence, and that it is hard to argue purely theoretically whether metacognition should be associated with a specific, dedicated neural basis. We are therefore happy to read that the reviewer believes the data reported in our paper can contribute to addressing these issues. As suggested, we now clarify in the revised manuscript that the psychological construct metacognition is used to explain behavioral findings like the accuracy of introspective confidence reports, but that it remains debated whether metacognition is uniquely implemented at the neural level. In our view, previous correlational imaging studies showing co-variation between neural activation and accuracy of confidence reports cannot convincingly resolve this debate, as they cannot show that brain activation (e.g., in frontopolar cortex) causally implements metacognitive processes. Our findings inform this debate by providing such a causal link between frontopolar cortex oscillations and metacognition.

In the introduction section, we clarify that it is still debated whether metacognition is a natural kind as well as how our study informs this debate (p.3):

“The psychological literature conceptualizes metacognition as a construct that allows explaining the accuracy of introspective confidence reports, but for a long time it has been debated whether metacognition represents also a natural kind at the neural level⁷. Previous research has documented a correlative link between metacognition in value-based choice and activity in frontopolar cortex (FPC)^{8,9}, but it remains unclear whether this FPC activity indeed causally contributes to choice-related confidence or whether it just relates to confidence without any behavioral implications. To decide whether FPC activity constitutes a functionally relevant neural substrate of metacognition, rather than just a correlate of behaviors associated with metacognition, it is necessary to show that modulating FPC excitability changes metacognitive confidence reports”

In the discussion section, we emphasize that our findings inform the debate on whether metacognition is a natural kind (p.11):

“Taken together, our study provides evidence that frontopolar theta oscillations constitute a neural substrate for metacognition, informing the debate on whether metacognition can be considered as a natural kind at the neural level.”

General Comments

In line with my comments in the preceding paragraph, I would urge the authors to consider a reorganization of the presentation into two experiments on metacognition and the FPC. They could start with a clear statement of the metacognitive hypothesis of the FPC and then work through their two experiments. I think this would come across as much less disjointed, and it would be clearer why these two experiments go together. As writing one kind of wonders if the experiments are being presented together simply because they were run together.

Response: We thank the reviewer for this helpful suggestion on how to streamline our manuscript. Following this advice, we now re-organized the presentation of the two experimental tasks in the introduction section in order to clarify the link between the confidence accuracy and the precommitment task. We explain that the confidence task and the precommitment task tap into two distinct aspects of metacognition: Metacognition about retrospective judgements (i.e., accuracy of confidence reports in choices that were just made) versus about prospective judgements (i.e., the ability to reliably predict future choices/behavior). Accordingly, we now re-named the “confidence task” to “confidence accuracy task” in order to emphasize that this task measures the accuracy of retrospective confidence judgements. If the FPC represents a unified neural substrate for metacognition, FPC tACS should improve both the accuracy in retrospective confidence judgements in the confidence accuracy task and the ability to reliably predict preference reversals in the precommitment task. Moreover, as the precommitment task provides only an implicit measure of prospective judgements, we also assessed whether metacognitive accuracy quantified by the explicit confidence reports in the confidence accuracy task predict the sensitivity to potential preference reversals when making prospective precommitment decisions.

In the revised manuscript, we describe the metacognitive link between the two experimental tasks on p.5:

“Metacognition is important in both retrospective and prospective judgements⁷. In retrospective judgements, metacognition is quantified as the accuracy of reported confidence in a choice made. Prospective judgements, in contrast, occur (often implicitly) prior to a decision or performance of a task, and metacognitive accuracy in prospective judgements indicates whether an individual can reliably forecast future decisions or task performance. If FPC theta oscillations constitute a neural substrate for

metacognition, theta tACS over FPC should improve metacognition in both retrospective judgements and prospective decision making. We therefore assessed the impact of FPC tACS on (1) a confidence accuracy task measuring metacognition as accuracy of retrospective confidence judgements and (2) a task measuring metacognition as capacity to assess the risk of preference reversal in prospective self-control decisions about whether to restrict one's access to future temptations. Lastly, because prospective self-control represents only an implicit indicator of metacognition, we also tested the hypothesis that metacognition quantified by explicit retrospective confidence judgements predicts individual differences in prospective self-control, as posited by formal models of precommitment^{12, 16}.”

Turning to the experimental results themselves, I think that the biggest disappointment is that while the theta stimulation does have a big effect, which is cool, so does gamma stimulation. The authors note appropriately that this is only significantly true in the first experiment, but to my eye it looks like its probably true in the second experiment, but that it does not rise to significance for reasons of inadequate power. That is really, to my mind, the greatest weakness of the paper. One really wants to show that increasing theta increase all of these metacognitive behaviors and that decreasing theta reduces these behaviors. That is really the result that the authors need to strive for in the future. One cannot help but wonder (and let me say that I am no tACS expert) whether there is a stimulation parameter set that would be better for disrupting theta than gamma? What if the tACS protocol had plenty of power at theta frequencies but either with coherent or scrambled phase, might that be a better approach? I'm sure the authors know better than I, but in the long run they will have to make this part of the argument more convincing.

Response: We agree with the reviewer that in our experiment, gamma tACS also increases metacognitive sensitivity relative to sham tACS (while in in the precommitment task there are no significant differences between gamma and sham). Choosing the optimal active control frequency is a tricky issue in tACS designs. We had chosen gamma as control frequency because a previous EEG study had provided no evidence for gamma involvement in metacognition (Wokke et al., 2017, *Journal of Neuroscience*). Accordingly, we did NOT expect a reduction of metacognitive sensitivity with gamma tACS, but this prediction did not hold at least for the current study design. One possible explanation for this effect may be phase

coupling between theta and gamma oscillations, with gamma rhythms nested in theta oscillations thought to be involved in transferring information across large spatial and temporal scales (Segneri et al., 2020, *Frontiers in Computational Neuroscience*). From this perspective, it seems possible that both theta and gamma oscillations are involved in metacognition, with theta being crucial for integrating information from other brain regions and gamma reflecting more local information processing. In any case, it is important to note that we find theta tACS to improve metacognition *not only* relative to sham *but also* relative to gamma tACS. Thus, regardless of any possible gamma effects, we can safely conclude that theta oscillations implement metacognition more strongly than other frequencies.

To more directly address this issue, we discuss the significant effect of gamma tACS on metacognition in the revised manuscript on p.10-11:

“Gamma tACS also increased metacognition relative to the sham condition but improvements in metacognition were significantly stronger for theta than for gamma stimulation. Gamma oscillations have previously been found to be nested in theta oscillation in several brain regions²⁶, and such cross-frequency couplings might be functionally relevant for transferring information across spatial and temporal scales²⁷. This notion further supports our interpretation that theta rhythms might implement metacognition via integration of information from distributed brain networks. Consistent with the idea of cross-frequency couplings, however, gamma oscillations too might be involved in metacognition, though on a different (more local) spatial scale and significantly weaker than theta oscillations.”

More Specific Comments

I thought that the most interesting and entirely unexplored result in the paper is the observation that the theta stimulation has a huge effect on variance. Just looking at figs 2 and 3 I was struck by how different the theta effect was in this regard. This stands out just from seeing how much bigger the difference is between the medians than between the means. I felt that this was a huge missed opportunity. In any revision the authors need a systematic exploration of this feature of the data. It probably has much to do with mismatches between their exact stimulation protocol and the intrinsic neural dynamics of the individual subjects. If some subjects have higher or lower theta, if phase is aligned or misaligned, if location of stimulation and location of the relevant theta are aligned, these things all probably control the efficacy of stimulation and their

data suggests that this is true – but there isn't a hint of this in the paper. I think that in any revision that simply has to be addressed.

Response: We thank the reviewer for pointing to this interesting aspect of our data. We agree that theta tACS indeed increases variation in individual coefficients, particularly for metacognitive sensitivity in the confidence accuracy task. There might be several reasons for this effect: As suggested by the reviewer, the stimulation frequency of 5 Hz might have aligned differentially with the phase of the individual frontopolar theta oscillations across participants. It has also been shown that the strength of tDCS effects depends on individual differences in neuroanatomy or localization of a cognitive function, as well as other factors like thickness of skull and cerebrospinal fluid (Opitz et al., 2015, *NeuroImage*). Independently of the variability of the strength of tACS effects, however, we think it is important to emphasize that this variability does not alter our main conclusion that in the mean theta tACS improves metacognition and sensitivity to preference reversals.

As suggested, we discuss explanations for the increased variance in metacognition under theta in the results section on p.8:

“Individual coefficients (Figure 2D) suggest that variation in metacognitive sensitivity was larger under theta compared with sham and gamma tACS. This pattern might result from variation in the degree of alignment between individual frontopolar theta rhythms and the applied stimulation frequency of 5 Hz, or from individual differences in the general susceptibility to brain stimulation²⁴, factors which add to the variation in baseline metacognitive sensitivity. This suggests interesting hypotheses for future experiments that may directly manipulate these factors. Irrespective of these considerations, our present results show that stimulation designed to enhance frontopolar theta oscillations indeed improves the ability to track objective decision uncertainty, supporting the view that these oscillations constitute a causal neural mechanism enabling metacognition of value-based choice processes.”

The comment above really also goes to the question of how best to analyze these data. With a linear regression approach as done here or with a more non-parametric approach. There are so many assumptions in the GLM and we have so little idea what is really going on with the

tACS. If the authors want a reader to stick with the GLM they should say a bit about the distributional structure of the data supporting this approach

Response: We thank the reviewer for bringing up this important issue. For the analysis of binary choice data, the use of mixed generalized linear models (MGLMs) has become the standard approach in decision neuroscience. MGLMs have several advantages over other approaches that analyze aggregated data with parametric or non-parametric tests. Because the binary choice data from all trials serve as the dependent variable in MGLMs, MGLMs account for the full variability in the data set, contrary to approaches based on aggregated mean data, which by definition underestimate the intra-individual variability due to the aggregation procedure. Moreover, to the best of our knowledge MGLMs for binary data make only minimal assumptions on the distribution of the dependent variable (contrary to mixed linear models with *continuous dependent variables*), as the only requirement is that the dependent variable follows a binary binomial distribution (which is by definition fulfilled for binary choice data). Note also that the independent (i.e., predictor) variables do *not* need to be normally distributed in MGLMs. We therefore believe that our approach is the best way to analyze the variability in our data set with almost no assumptions on the distribution of the dependent and independent variables.

We motivate our choice of generalized mixed linear models in the revised manuscript on p.17:

“The advantage of MGLMs over other statistical procedures is that MGLMs provide a better account of the full variation in choice data sets with binomially distributed binary dependent variables, compared to participant-specific aggregated approaches that neglect intra-individual variability on the trial level. In MGLMs statistical inferences are based on group-level fixed effect estimates while accounting for inter-individual variation via random effects.”

At a more methodological level I also had one small question: How were the subjects actually paid in experiment 2? Paying out Smaller-sooner vs Larger-later experiments is annoying and important and I just wanted to understand if the same mechanism was used for now, 28d and longer delays, and what was that mechanism.

Response: Whenever an amount in the future was selected for payment (independently of whether this was the case in the confidence accuracy or the precommitment task, or after 28 days versus longer delays), the amount of money was sent to participants via mail. We added a clarifying statement in the Materials and Methods section (p.16):

“If participants chose an option to be paid out after the experiment (either in the confidence accuracy or in the precommitment task), the given amount was sent to the participant via mail.”

Reviewer #3:

Remarks to the Author:

In this study the authors use tACS to investigate the relationship of frontopolar theta oscillations and metacognition of economic preference. The authors came up with an interesting design to address this timely topic in the field of human decision-making. The results could be potentially interesting and provide us with additional insight into the mechanisms underlying our ability to control our own decision process. Below you will find my main comments:

Response: We thank the reviewer for evaluating our study as addressing a timely topic and as providing interesting insights.

Throughout the manuscript the authors claim that the main effects are related to increased theta power in frontopolar cortex. However, the spatial specific claims seem to be unwarranted with the current methods used (i.e., not using HD-tDCS). The strength of tACS might also not be in spatial specificity but in enhancements of coherence between brain areas. In addition, the plot showing the modeling of the electric field density also shows large spread to DLPFC.

Response: We agree that tACS does not allow strong claims regarding local specificity. In the discussion section, we accordingly added a caveat clarifying that besides frontopolar cortex our stimulation protocol might have affected also other regions like dorsomedial prefrontal cortex, VMPFC, or DLPFC. While parts of the dorsomedial prefrontal cortex also appear involved in metacognitive processing (Vaccaro and Fleming, 2018), VMPFC and DLPFC have been associated more strongly with confidence and self-control in hyperbolic discounting. We argue that the lack of significant stimulation effects on confidence and self-control (indicated by discount parameters k) therefore speaks against potential tACS effects on these regions, though we cannot completely rule out this possibility (p.12).

“We also note that due the relatively low spatial specificity of tACS, we cannot rule out that the observed results reflect stimulation effects on brain regions adjacent to FPC, including dorsomedial prefrontal cortex, VMPFC, and DLPFC. However, our current findings provide no evidence for tACS effects on confidence ratings or the degree of hyperbolic discounting, aspects of behavior that have been linked to VMPFC and

DLPFC^{8, 11, 35}. Thus, stimulation effects on FPC provide a more parsimonious account of our data, but it remains to be seen whether the stimulation effects concern local activity or coherence of FPC communication with other brain regions.”

Some of the main findings have p-value of $p=0.04$ (precommitment results) and $p=0.049$ (theta gamma difference in the confidence task). It is a bit unclear at the moment how stable these effects are. Maybe a bootstrap procedure could help with this.

Response: We thank the reviewer for this helpful comment that allowed us to underline the robustness of our findings. Following the suggestion of the reviewer, we now test the significance of fixed effects via bootstrap instead of the Wald tests we reported in the previous version of the manuscript. For these analyses, we determine the 95% confidence intervals for all fixed effects using the bootMer function in R. For all effects showing a significant p-value below 5% (including the ones close to the statistical threshold mentioned by the reviewer) the 95% confidence interval does not include zero, supporting the significant results from the p-values. In the manuscript (main text and Tables 1 and 2) we now report the borders of the 95% confidence intervals instead of p-values from the Wald statistics to assess the significance of fixed effects.

We describe the bootstrap procedure in the revised manuscript on p.17:

“We assessed the significance of fixed effects by the 95% confidence intervals ($CI_{\text{bootstrap}}$) determined via parametric bootstrap (implemented by the bootMer function in R), which provides more reliable results than p-values based on Wald statistics.”

The individual time preference itself can also be changed by stimulation. Each individual's time preferences is fitted by a hyperbolic discount function to the choices in the confidence task. However, it is unclear what data has been used for the fit (e.g., all data, or sham data) and whether potential changes in preference due to stimulation occur.

Response: In the revised manuscript, we now clarify that for each participant we estimated a discount parameter k separately for each stimulation condition and that discount parameters did not significantly differ between stimulation conditions (p.6):

“To test our hypotheses, we first determined the subjective value of each choice option by fitting hyperbolic discount function to the individual choice data, separately for each tACS condition (see Materials and Methods). Discount parameters k did not differ significantly between stimulation conditions, Wilcoxon rank sum tests, all $W < 651$, all $p > 0.72$.”

In the Materials and Methods section, the model estimation approach is now described as follows (p.17):

“For that purpose, we first estimated each individual’s time preferences by fitting a hyperbolic discount function to the choices in the confidence accuracy task, separately for each tACS condition (equation 1)”

How is the z-transform performed on the confidence scores? If this has been done per participant across conditions, it could be that specific stimulation protocols affected response bias instead of metacognitive sensitivity (overall higher/lower confidence reports irrespective of DV), see Masson, M. E. J., and Rotello, C. M. (2009).

Response: We thank the reviewer for this insightful comment. As described in the manuscript, the z-transformation of confidence score had been done per participant across condition. As we now report on p.6, there were no significant differences in confidence ratings between tACS conditions, but we agree that it is important to rule out the possibility that these non-significant effects on confidence might have affected the results for metacognitive sensitivity. We therefore performed a control analysis where we additionally subtracted the mean confidence rating in each tACS condition from the confidence scores. This control analysis replicated the significant effects of theta tACS on metacognitive sensitivity relative to sham and gamma tACS, providing evidence for the robustness of our findings (p.7).

“We note that this result pattern was robust to controlling for potential tACS effects on confidence (which were non-significant, as shown above) by subtracting the mean confidence rating in each tACS condition from confidence scores (tACS_{theta-sham} × DV × Confidence, beta = 2.38, CI_{bootstrap} = [1.44; 4.06]; tACS_{theta-gamma} × DV × Confidence, beta = 1.36, CI_{bootstrap} = [0.38; 2.81]).”

Recently there are many interesting findings with respect to gamma nested in theta. By using gamma as an active control causes some issues with respect to these observations. In case theta serves communication between brain areas by nesting gamma bursts in each cycle, the present finding of a significant gamma effect might support such a mechanism.

Response: We thank the reviewer for this insightful comment. In the revised discussion section, we clarify that couplings between theta and gamma oscillations might be involved in transferring information across brain networks, which appears consistent with our interpretation that theta oscillations implement metacognition by integrating information from different brain regions (p.10-11).

“Gamma tACS also increased metacognition relative to the sham condition but improvements in metacognition were significantly stronger for theta than for gamma stimulation. Gamma oscillations have previously been found to be nested in theta oscillation in several brain regions²⁶, and such cross-frequency couplings might be functionally relevant for transferring information across spatial and temporal scales²⁷. This notion further supports our interpretation that theta rhythms might implement metacognition via integration of information from distributed brain networks. Consistent with the idea of cross-frequency couplings, however, gamma oscillations too might be involved in metacognition, though on a different (more local) spatial scale and significantly weaker than theta oscillations.”

FPC has many other functions that could be relevant in the present design. Some of them: the formation of internal predictions and forecasting, and the “somatic marker” hypothesis (see for instance Momennejad - Current Opinion in Behavioral Sciences, 2020; Wokke & Ro - Journal of Neuroscience, 2019, T Poppa, A Bechara - Current opinion in behavioral sciences, 2018).

Response: We fully agree with the reviewer that FPC has also been linked with other functions than metacognition, including internal predictions and forecasting as well as the somatic marker hypothesis. We included the suggested references in the discussion section and clarify that the FPC’s precise role in cognition has not been clarified yet (p.12).

“We note that besides metacognition the FPC has also been related to other functions that might promote precommitment and future-oriented behavior, for example representing predictive cognitive maps^{31, 32, 33} or integrating interoceptive signals into goal-directed behavior³⁴. The FPC’s precise role in cognition is still controversially debated, and it is currently unclear how different views can be integrated. In the context of the current study, however, it seems most parsimonious to explain the stimulation effects on precommitment via the FPC’s role for metacognition, given the FPC’s well established role for metacognition in the literature⁹ as well as the importance for metacognition for precommitment¹⁷.”

The design assumes that the LL is the better option. However, this all depends on the participants. In case they are students knowing that in 28 days it’s the end of the month and they need the money, the value parameters change. Therefore to postpone the decision and see how the financial situation is at the moment could in some instances be the better option, depending on circumstances. Alternatively, participants might choose the option “not to choose again”. Having nothing to do with the actual value.

Response: We fully agree with the reviewer that precommitting to the LL reward might not have been the optimal choice for all participants depending on the individual financial situation and the timing of the late payment relative to other income streams. While it is plausible that individual differences in the financial situation increased the inter-individual variance in participants’ choices (note that we controlled for such individual differences by modelling random slopes), our results nevertheless suggest that across all participants, higher metacognitive sensitivity is linked with precommitment choices. We discuss this caveat in the revised manuscript on p.11:

“While our findings support the view that metacognition moderates precommitment decisions, we acknowledge that precommitting to the LL reward might not be the optimal choice for all individuals, e.g. for individuals in difficult financial situations. Although individual differences in the financial situation are plausible to have increased variance in participants’ behavior, we note that across all participants metacognitive sensitivity predicts higher willingness to precommitment to LL options.”

What does “this effect was numerically stronger” mean?

Response: We apologize for this unclear wording. With the parametric bootstrap procedure suggested by the reviewer, this effect is significant, and we re-formulated this sentence as follows (p.8-9):

“Again in line with our hypotheses, theta tACS increased the sensitivity of precommitment choices to potential preference reversals relative to sham, $tACS_{\text{theta-sham}} \times \text{Reversal risk}$, $\beta = 2.05$, $CI_{\text{bootstrap}} = [0.04; 6.24]$ (Figure 3A/B and Table 2); this effect was stronger when participants currently preferred the LL over SS reward and might thus reverse their preference from the LL to the SS reward at the time of the final choice, $tACS_{\text{theta-sham}} \times DV_{\text{initial}} \times \text{Reversal risk}$, $\beta = 2.02$, $CI_{\text{bootstrap}} = [0.45; 4.20]$.”

Reviewer #4

In the manuscript by Soutschek et al. the authors explore the effect of theta tACS on choice behavior.

Below I focus on several methodological aspects of the study. Overall, there seems to be only weak support that tACS affected task performance in a meaningful and specific manner.

Response: We thank the reviewer for the insightful comments on the manuscript. As outlined below, we provide additional analyses and discuss the specificity of our stimulation protocol based on the reviewer's comment, which improved the quality of the manuscript.

1. The authors explored two active stimulation conditions (plus sham) during two tasks, both with multiple readouts. The statistical significance of the tACS effect was established using a series of linear models (22 reported regressors between Tables 1 and 2). However, I don't see any attempts to correct for the multiple comparisons that arose from so many statistical models in the paper.

Response: We thank the reviewer for raising this important issue. First, it is important to emphasize that we did not “explore two active stimulation conditions”. In contrast, in our models we tested the theory-guided *a priori* predictions that theta tACS, relative to sham and an active control frequency, is causally involved in reporting decision confidence and in precommitment. As stated in the manuscript, the hypothesis for theta oscillations is based on a previous EEG study relating metacognition to frontopolar theta oscillations (Wokke et al., 2017, *Journal of Neuroscience*). As this study provided no evidence for gamma involvement in metacognition, gamma tACS was selected as an active control frequency.

Second, we note that statisticians disagree on when Bonferroni corrections should be used (Perneger, 1998, *BMJ*). Advocates of Bonferroni corrections argue they should be used if one hypothesis is tested by several tests and if a significant result in *only one of the performed tests* is considered as support for the hypothesis (Cabin & Mitchell, 2000, *Bulletin of the Ecological Society of America*; Rice, 1989, *Evolution*). Importantly, this is not what is done in the current study. We test the hypothesis that metacognition is improved by theta tACS, relative to both sham and gamma tACS. That is, we consider our hypothesis as supported *only if both* of these

tests (and not just *one* of them) show a significant result. In contrast, Bonferroni corrections would be necessary if a significant result in only one of these tests was considered as evidence for our hypothesis or for cases of multiple exploratory post-hoc tests. Thus, in our study the tests for theta versus sham and theta versus gamma are used to provide robust and converging evidence that theta tACS affects metacognition, relative to two control conditions. Given that two statistical tests need to be significant to support the hypothesis, it would even be justified to increase (instead of to reduce, as in Bonferroni corrections) the alpha levels of the single tests in order to achieve an overall alpha of 5% for hypothesis testing (<https://daniellakens.blogspot.com/2016/02/why-you-dont-need-to-adjust-your-alpha.html>). In any case, it should become clear that correcting for multiple comparisons is not necessary here because not just one significant result would be sufficient to reject the null hypothesis, which is the crucial assumption underlying Bonferroni correction.

Third, the reviewer seems to worry about the total number of predictors in the statistical model, but here our apriori hypotheses require that very specific fixed effects show significant results (for example, the hypothesis that theta tACS improves metacognition requires both the “tACS_{theta-sham} × DV × confidence” and “tACS_{theta-gamma} × DV × confidence” interactions to be significant). Again, significant results for any other fixed effects (e.g., “tACS_{theta-gamma} × DV”) would not be regarded as evidence for the hypothesis that theta tACS improves metacognition. Because correction for multiple comparisons applies only to the statistical comparisons testing one hypothesis (Cabin & Mitchell, 2000), additional predictors that are irrelevant for the given hypothesis do not imply the need for corrections for multiple comparisons. Finally, also the two models reported in Tables 1 and 2 test two separate hypotheses in two distinct tasks, namely whether theta tACS affects retrospective metacognition (confidence accuracy task) or the influence of reversal risk on prospective precommitment decisions (precommitment task). Again, since corrections for multiple comparisons are only necessary if one hypothesis is tested by several independent tests, we do not need to apply it for tests of two distinct apriori hypotheses with two different models.

In the revised manuscript, we explain in more detail why no Bonferroni correction for the theta versus sham and theta versus sham comparisons are necessary (p.17):

“Importantly, our hypothesis that theta tACS improves metacognition is only supported by significant results for the comparisons between theta tACS and *both* sham tACS (passive control) and gamma tACS (as active control), and not by significant results for

just one of these comparisons. Alpha correction for multiple comparisons was therefore not required⁴⁰.”

2. The main effect of the theta stimulation itself is not significant in either task ($p=0.94$ in the confidence task; $p=0.08$ in the precommitment task, both not corrected) and only reaches significance in interaction with other statistical factors (choice confidence, subjective value difference, or reversal risk). Importantly, these other factors are not the same between the two tasks, which further complicates the interpretation.

Response: We thank the reviewer for this comment. It is true that the main effects of tACS are not significant in the mixed models, but these effects are irrelevant for our hypotheses (as explained in our response to the previous comment). In the confidence accuracy task, a significant main effect of tACS would indicate that tACS leads to more/less choices of larger-later versus smaller-sooner rewards in general, which is completely different from tACS effects on metacognition. Indeed, our hypothesis was not that the frontal pole is causally involved in patience, but that it plays a crucial role in metacognition as indexed by the interaction between tACS, value difference, and confidence ratings (note that the value difference \times confidence interaction is routinely used as index of metacognition in the literature). Similarly, the lack of significant tACS main effects in the precommitment task just shows that tACS does not change precommitment choices per se, which is irrelevant for our hypothesis that metacognition changes precommitment decisions when the risk of preference reversals is high. This specific hypothesis needs to be tested by the tACS \times Reversal risk \times DV_{initial} interaction, as implemented in the manuscript. In the revised manuscript, we have now clarified this rationale (p.18 and 19 to prevent potential misunderstandings in future readers.

“Assessing the impact of tACS on the DV \times Confidence interaction effect thus allowed us to test whether stimulation modulated the degree to which participants were metacognitively aware of their time preferences.”

“If theta tACS increases the sensitivity to potential preference reversals, this should be expressed by significant tACS effects on the Reversal risk \times DV_{initial} interaction.”

Furthermore, the two statistical models entail different predictors because they analyze behavior in two distinct tasks assessing separate psychological constructs with different dependent variables. The confidence accuracy task measures the metacognitive ability to reliably report decision noise, and the model for this task therefore includes predictors for value difference (to assess decision noise) and reported confidence. Likewise, as the precommitment task measures determinants of the willingness to precommit, the statistical model includes predictors for the risk of preference reversals when having to make the final choice. Thus, the structure of the administered tasks does not allow analyzing them with the exactly identical models (there are no confidence ratings in the precommitment task, and it is not possible to compute the variable reversal risk in the confidence accuracy task), and even if it was possible it would not appear meaningful. However, note that all task-unspecific predictors (tACS, discomfort ratings, task order) are in fact identical in the models. We make these points explicit in the revised manuscript (p.19 so that future readers understand this rationale.

“Thus, the MGLM for the precommitment task included the same predictors as the MGLM for the confidence accuracy task, except that we replaced the predictor for confidence ratings (which were not measured in the precommitment) with a predictor for reversal risk, as the goal of the precommitment task was to measure the willingness to precommit as function of the risk of preference reversals.”

3. Although the authors performed a computational model of the stimulation, this is not well described in the manuscript. I can only see one corresponding visualization (Figure 1, panel D). For some reason, the values on the figure are restricted between 0.1 and 0.2 V/m, which are probably the half-maximum and maximum values. Such partial presentation and lack of any quantitative analysis give little confidence in what brain areas were actually stimulated. In addition, there is no control stimulation montage (location) employed in the study. So, one cannot confidently attribute any stimulation effects to a specific brain area.

Response: We thank the reviewer for this comment. In the revised manuscript, we have now changed the value range in Figure 1D from 0 to 0.2 V/m (maximum value). We agree with the reviewer that due to the relatively low spatial resolution of tDCS, one needs to be careful with drawing strong inferences on which brain regions were stimulated. We therefore added a caveat to the discussion section clarifying that the stimulation might have affected also brain regions

adjacent to frontal pole, including dorsomedial prefrontal cortex, dorsolateral prefrontal cortex, and ventromedial prefrontal cortex. We also emphasize that, following common approaches in tACS research, we employed only a control frequency but no control stimulation site, such that we cannot draw strong inferences about the local specificity of the observed effects (p.12-13).

“We also note that due the relatively low spatial specificity of tACS, we cannot rule out that the observed results reflect stimulation effects on brain regions adjacent to FPC, including dorsomedial prefrontal cortex, VMPFC, and DLPFC. However, our current findings provide no evidence for tACS effects on confidence ratings or the degree of hyperbolic discounting, aspects of behavior that have been linked to VMPFC and DLPFC^{8, 11, 35}. Thus, stimulation effects on FPC provide a more parsimonious account of our data, but it remains to be seen whether the stimulation effects concern local activity or coherence of FPC communication with other brain regions. Finally, it is worth mentioning that our experimental design did not include an active control region, which further underlines that we cannot draw strong conclusions regarding the local specificity of the observed stimulation effects.”

4. One major confound that could have impacted the results here are phosphenes (activation of the eye's retina by the alternating currents). The authors show no results of post-stimulation questionnaires regarding the presence of phosphenes, although the stimulation electrodes were right above the eyes.

Response: We agree with the reviewer that in tACS protocols where one electrode is close to the eyes, phosphenes are a potential issue. In the revised manuscript, we now report the results from post-stimulation questionnaires in which participants rated the strength of the perceived phosphenes as well as their impact on task performance on Likert scales from 1 to 9. On average, participants indicated that they had perceived phosphenes (mean rating = 5.6), but these phosphenes had next to no impact on task performance (mean rating = 2.9). To control for potential influences of phosphenes (operationalized as “flickering” in the questionnaire) and discomfort on task performance, we now added these ratings as covariates of no interest to the statistical models, and mention that adding these covariates does not change the pattern of results (p.16).

“At the end of the experiment, participants indicated the tDCS-induced discomfort, whether they perceived flickering during tACS, as well as whether the discomfort or flickering affected their task performance on Likert scales from 1 to 9. The mean ratings reported for discomfort and flickering were 4.1 and 5.6, respectively, but participants reported only low to moderate disturbing influences of discomfort (mean = 3.4) and flickering (mean = 2.9) on task performance. In order to control for any influences of tACS-induced irritations on task performance, we added the individual ratings for the impact of discomfort and flickering on task performance as control variables to all statistical models.”

5. Despite the rather complex nature of the cognitive task, which included a choice and a confidence rating, the authors opted to use a very limited number of task trials - 30 or 60 trials per stimulation condition in the precommitment or confidence tasks, respectively. That is most likely not enough to reliably estimate individual parameters of metacognitive effects using a Bayesian approach.

Response: We thank the reviewer for raising this issue. First, it is important to note that our inferences on the effects of tACS in the precommitment and confidence accuracy tasks are not based on individual parameter estimates. Instead, we analyzed data with hierarchical generalized mixed linear models, in which statistical inferences are based on the group-level fixed effect estimates (while accounting for individual variation via random effects). Thus, the reliability of individual parameter estimates is not crucial for our group-level based inferences regarding the effects of theta tACS on precommitment and metacognition. In both tasks, the crucial parameters were estimated based on a substantial number of trials: 6660 trials for the confidence accuracy task, 3330 trials in the precommitment task. In the revised manuscript, we clarify that in the applied mixed linear models, the statistical inferences are based on the group-level, not the participant-level, effects (p.17):

“In MGLMs statistical inferences are based on group-level fixed effect estimates while accounting for inter-individual variation via random effects.”

Second, the Bayesian estimation of discount parameters was performed only in the confidence accuracy task, i.e., on the basis of 60 trials. Such a number of trials is usually considered as

sufficient for estimating discount parameters, and numerous studies use even less trials for parameter estimation (Ballard et al., 2017, *Psychological Science*; Figner et al., 2010, *Nature Neuroscience*; Jenkins & Hsu, 2017, *Psychological Science*; Reeck et al., 2017, *PNAS*). Moreover, parameter estimates in the three tACS conditions were highly correlated, underlining the reliability of individual parameter estimates, all $r > 0.95$, all $p < 0.001$. Finally, all chains in the Bayesian estimation procedure converged (indicated by \hat{R} values below 1.01), which also underlines the robustness of parameter estimates. In the revised manuscript, we provide more details regarding the Bayesian parameter estimation procedure on p.17-18:

“All chains converged, as indicated by \hat{R} values below 1.01. Moreover, discount parameters in the three tACS conditions were strongly correlated with each other, all $r > 0.95$, all $p < 0.001$, providing evidence for the reliability of individual parameter estimates.”

REVIEWER COMMENTS

Reviewer #1 (Remarks to the Author):

The authors have comprehensively responded to my previous comments, and I was glad to see the new control analyses examining tACS effects on confidence level

Reviewer #2 (Remarks to the Author):

I found the revised version of the manuscript to be very much improved. The authors have addressed all of the comments from my review and from the thoughtful reviews by #1 and #3. The manuscript is now a very clear and sharp statement of their hypotheses and demonstrates a successful test of those hypotheses. I believe that the paper is now a great addition to the literature.

Reviewer #3 (Remarks to the Author):

I'm satisfied with the revised manuscript. The authors addressed all my comments and nuanced the interpretation of the findings accordingly.

Reviewer #5 (Remarks to the Author):

Remarks to the authors (numbered according to the comments of reviewer #4):

1. Correction for multiple comparisons due to many statistical models

The authors explain convincingly that they have used one model for each experiment and why they used the factors.

2. Main effect not significant

I agree with the authors that it is sufficient for their claims to demonstrate significance of the mentioned interactions rather than the main effect.

3. Computational model of brain stimulation

Fig. 1 D looks convincing now.

Another aspect of the description of the electrical stimulation, however, is confusing. On page 16, the authors state: 'We applied tACS using two 8-channel tDCS stimulators (DC-stimulator MC, neuroConn, Ilmenau, Germany).' Later, it is mentioned that the two EEG electrode sites Fpz and CPz were used for stimulation. How can only two electrodes be connected to two stimulators with eight channels each, i.e. 16 wires. I suppose there is an error in the description.

It should also be noted as a limitation that very short duration were used for stimulation blocks (3 minutes each). Most previous studies used significantly longer durations (e.g. 20 to 60 minutes). It has been argued in the literature for tDCS and tACS, that – at least for after-effects to occur – longer durations are probably required.

4. Phosphenes

It is a pity that the authors did not control in a better fashion for the potential effects of phosphenes. It has been argued repeatedly that phosphenes could be an alternative explanation of tACS effects. This should be discussed and the relevant articles should be cited. In the literature on phosphenes, it is discussed that the influence of phosphenes on subjects' performance need not be conscious. Therefore, the rating of the participants whether they believe that the phosphenes influenced their behavior is not very meaningful. The effects of the manuscript might be solely explained by the phosphenes. This limitation clearly challenges the strength of the manuscript.

5. Statistics / number of trials

The authors mention that the parameters for statistics were computed for many thousand trials. This is convincing.

All statistical testing, however, is based on parameters that are derived from 'fitting hyperbolic discount functions' to individual data. From Fig. 3, it becomes obvious that the three conditions yielded very different fits. It needs to be described whether the three models were identical, which parameters of the functions were fixed and which ones were fitted. In addition, individual data and a measure of their variance (i.e. standard error of the mean) need to be plotted in Figures 2 and 3. At the moment, there is no way for the reader to evaluate how well the model fits the data. In my opinion, the model is a sigmoid function not a hyperbolic one.

I guess, equation 1 was used to generate Fig. 2 and equation 2 was used to generate Fig. 3. If this is correct, it should be mentioned in the manuscript. I was surprised that the axes of the figures were not identical to the variables in the equations. There should be a one-to-one correspondence in order to understand the relation of the figure to the equations.

Apart from the previous comments of reviewer #4, I find the hypotheses rather exploratory.

On page 5, the authors state:

[We] 'tested whether this stimulation (compared to neural-ineffective control stimulation) indeed affects metacognitive judgements.'

'we also tested the hypothesis that metacognition quantified by explicit retrospective confidence judgements predicts individual differences in prospective self-control, as posited by formal models of precommitment'

A clear hypothesis should read as follows: We hypothesize that theta-tACS will increase (or decrease) parameter X of our paradigm. In contrast to (or in line with) the first hypothesis, gamma-tACS will decrease (or increase) parameter X of our paradigm.

REVIEWER COMMENTS

Reviewer #1 (Remarks to the Author):

The authors have comprehensively responded to my previous comments, and I was glad to see the new control analyses examining tACS effects on confidence level

Response: We are happy to hear that the reviewer was satisfied with our responses.

Reviewer #2 (Remarks to the Author):

I found the revised version of the manuscript to be very much improved. The authors have addressed all of the comments from my review and from the thoughtful reviews by #1 and #3. The manuscript is now a very clear and sharp statement of their hypotheses and demonstrates a successful test of those hypotheses. I believe that the paper is now a great addition to the literature.

Response: We thank the reviewer for considering our paper as a great addition to the literature.

Reviewer #3 (Remarks to the Author):

I'm satisfied with the revised manuscript. The authors addressed all my comments and nuanced the interpretation of the findings accordingly.

Response: We are happy to hear that the reviewer was satisfied with our responses.

Reviewer #5:

Remarks to the authors (numbered according to the comments of reviewer #4):

1. Correction for multiple comparisons due to many statistical models

The authors explain convincingly that they have used one model for each experiment and why they used the factors.

Response: We thank the reviewer for judging our statistical analyses as convincing.

2. Main effect not significant

I agree with the authors that it is sufficient for their claims to demonstrate significance of the mentioned interactions rather than the main effect.

Response: We thank the reviewer for the positive evaluation of our response to reviewer 4's comment.

3. Computational model of brain stimulation

Fig. 1 D looks convincing now.

Another aspect of the description of the electrical stimulation, however, is confusing. On page 16, the authors state: 'We applied tACS using two 8-channel tDCS stimulators (DC-stimulator MC, neuroConn, Ilmenau, Germany).' Later, it is mentioned that the two EEG electrode sites Fpz and CPz were used for stimulation. How can only two electrodes be connected to two stimulators with eight channels each, i.e. 16 wires. I suppose there is an error in the description. It should also be noted as a limitation that very short duration were used for stimulation blocks (3 minutes each). Most previous studies used significantly longer durations (e.g. 20 to 60 minutes). It has been argued in the literature for tDCS and tACS, that – at least for after-effects to occur – longer durations are probably required.

Response: We apologize for this confusion. Of course the two electrodes on participants' heads were connected to only two of the channels of the tDCS stimulator. We note that multi-channel stimulators allow varying the number of stimulated subjects, such that not all channels need to

be used in an experimental session (i.e., an 8-channel stimulator does not imply that eight wires or electrodes are used in all sessions). The reason why we had used two stimulators is that we had tested several participants in parallel, and using two instead of one stimulator allowed us to run a higher number of participants simultaneously. However, we fully understand that mentioning “two 8-channel tDCS stimulators” might be confusing for readers (and unnecessary in the first place), and therefore we re-formulated this sentence as follows (p.16):

“We applied tACS using an 8-channel tDCS stimulator (DC-stimulator MC, neuroConn, Ilmenau, Germany).”

We also agree with the reviewer that most tACS studies employed longer stimulation durations. However, as we now discuss in the revised manuscript, this might be a strength rather than a weakness of our study. The reviewer states that according to the literature longer durations than 3 min are required to induce after-effects for offline tACS. Crucially, we were interested in effects of *online* (concurrent) rather than offline stimulation, such that the short stimulation durations allowed us to minimize the risk of stimulation after-effects and possible carry-over effects on the next experimental block. In the revised manuscript, we now clarify that the stimulation durations allowed us to minimize carry-over effects between the stimulation blocks on p.16:

“Thus, the total stimulation duration in each miniblock was 180 s, which allowed us to minimize the risk of stimulation-induced physiological after-effects.”

4. Phosphenes

It is a pity that the authors did not control in a better fashion for the potential effects of phosphenes. It has been argued repeatedly that phosphenes could be an alternative explanation of tACS effects. This should be discussed and the relevant articles should be cited. In the literature on phosphenes, it is discussed that the influence of phosphenes on subjects' performance need not be conscious. Therefore, the rating of the participants whether they believe that the phosphenes influenced their behavior is not very meaningful. The effects of the manuscript might be solely explained by the phosphenes. This limitation clearly challenges the strength of the manuscript.

Response: We thank the reviewer for this comment. While we controlled for perceived flickering in our statistical analyses, we agree with the reviewer that tACS-induced phosphenes might affect behavior without participants' awareness. In our study, we had therefore implemented the following procedures to reduce the potential impact of phosphenes: First, we performed extensive pilot studies to minimize phosphenes and identify an optimal control frequency with a similar amount of phosphenes. The literature suggests that phosphenes are most pronounced for oscillations in the beta and alpha range (Bland & Sale, 2019, *Experimental Brain Research*). We therefore decided for upper gamma as control frequency, as both theta and gamma tACS induced only weak or no phosphenes in our pilot experiment. Based on the pilot studies, we also maximized the light in the lab during the main experiment as the pilot study showed that increased light minimizes the perception of flickering. Second, while previous studies report effects of tACS-induced phosphenes on task performance (e.g., Spaak et al., 2014, *Journal of Neuroscience*; Schutter, 2016, *NeuroImage*), this seems to be the case mainly for visual perception tasks where visual acuity is key. It seems much less likely that phosphenes would affect value-based decision making. We could not find literature on (and cannot think of) a plausible mechanism by which tACS-induced phosphenes should improve value-based decision making; if anything, we would expect phosphenes to disturb rather than to improve decision making. However, we are open to incorporate any suggestions if the reviewer should know of such a mechanism.

In the revised manuscript, we clarify that we had chosen gamma as control frequency to match phosphenes with the active theta tACS condition. We also note that while we cannot logically rule out that tACS-induced phosphenes may have contributed to the observed result pattern, phosphenes appear to affect performance mainly in visual perception tasks (p.16):

“The control frequency of 80 Hz was determined in pilot experiments to match the tACS-induced discomfort and phosphenes between theta and control tACS. We note that phosphenes appear to affect performance mainly in visual perception tasks^{40, 41} and it seems much less likely that phosphenes would affect (and in fact improve) value-based decision making, but we cannot logically rule out this possibility.”

5. Statistics / number of trials

The authors mention that the parameters for statistics were computed for many thousand trials. This is convincing.

All statistical testing, however, is based on parameters that are derived from 'fitting hyperbolic discount functions' to individual data. From Fig. 3, it becomes obvious that the three conditions yielded very different fits. It needs to be described whether the three models were identical, which parameters of the functions were fixed and which ones were fitted. In addition, individual data and a measure of their variance (i.e. standard error of the mean) need to be plotted in Figures 2 and 3. At the moment, there is no way for the reader to evaluate how well the model fits the data. In my opinion, the model is a sigmoid function not a hyperbolic one.

I guess, equation 1 was used to generate Fig. 2 and equation 2 was used to generate Fig. 3. If this is correct, it should be mentioned in the manuscript. I was surprised that the axes of the figures were not identical to the variables in the equations. There should a one-to-one correspondence in order to understand the relation of the figure to the equations.

Response: We apologize for this misunderstanding. Figures 2 and 3 do not show the results of the fitted hyperbolic discount functions but the results of the mixed generalized linear model for the confidence accuracy task (MGLM-1) and the precommitment task (MGLM-2), respectively. As the MGLMs analyze binary choice data, Figures 2A-C and 3A indeed show sigmoid functions, because this is the link function used in MGLMs for binary dependent variables. It is therefore a misunderstanding that equations 1 and 2 were used to generate Figures 2 and 3. Instead, equations 1 and 2 were used to compute individual hyperbolic discount parameters (using the identical approach for all tACS conditions, and all parameter estimates converged, as described on p.18). The discount parameters in turn allowed us to compute the predictors "difference in value (DV)" that are included in MGLM-1 and MGLM-2 (as we describe on p.7 and p.18). We hope that this explains why the axes in Figures 2 and 3 do not match the parameters in equations 1 and 2, but instead correspond to the predictors included in MGLM-1 and MGLM-2, respectively. We note that Figures 2 and 3 also represent the variance in the different tACS conditions, as Figures 2D and 3B show boxplots for individual parameter estimates, illustrating the variance in parameter estimates via box and whisker length.

To avoid this misunderstanding, we added clarifying statements to the figure legends:

"Figure 2. Stimulation effects on metacognition based on the results of the MGLM for the confidence accuracy task."

"Figure 3. Stimulation effects on precommitment (based on the results of the MGLM for the precommitment task) and relation to individual differences in impulsiveness."

Apart from the previous comments of reviewer #4, I find the hypotheses rather exploratory.

On page 5, the authors state:

[We] ‘tested whether this stimulation (compared to neural-ineffective control stimulation) indeed affects metacognitive judgements.’

‘we also tested the hypothesis that metacognition quantified by explicit retrospective confidence judgements predicts individual differences in prospective self-control, as posited by formal models of precommitment’

A clear hypothesis should read as follows: We hypothesize that theta-tACS will increase (or decrease) parameter X of our paradigm. In contrast to (or in line with) the first hypothesis, gamma-tACS will decrease (or increase) parameter X of our paradigm.

Response: We thank the reviewer for these helpful comments and formulated directed hypotheses.

p.4:

“We thus applied a transcranial alternating current stimulation (tACS) protocol designed to enhance theta-band oscillations in the FPC and tested whether this stimulation (compared to neural-ineffective control stimulation) indeed improves metacognitive judgements.”

p.5:

“we also tested the hypothesis that higher metacognitive skills quantified by explicit retrospective confidence judgements predict better prospective self-control, as posited by formal models of precommitment^{12, 16}.”

REVIEWER COMMENTS

Reviewer #5 (Remarks to the Author):

The authors have addressed all of my concerns convincingly.